# VISUAL SYMBOLIC MECHANISMS: EMERGENT SYMBOL PROCESSING IN VISION LANGUAGE MODELS

**Rim Assouel[1*]    Declan Campbell[2*]    Yoshua Bengio[1,3]    Taylor Webb[1]**

[*] **Equal contribution.**
[1] Mila - Québec AI Institute, Université de Montréal
[2] Princeton Neuroscience Institute
[3] CIFAR AI Chair
assouelr@mila.quebec, idcampbell@princeton.edu,
yoshua.bengio@mila.quebec, taylor.w.webb@gmail.com

## ABSTRACT

To accurately process a visual scene, observers must bind features together to represent individual objects. This capacity is necessary, for instance, to distinguish an image containing a red square and a blue circle from an image containing a blue square and a red circle. Recent work has found that language models solve this 'binding problem' via a set of symbol-like, content-independent indices, but it is unclear whether similar mechanisms are employed by Vision Language Models (VLMs). This question is especially relevant, given the persistent failures of VLMs on tasks that require binding. Here, we identify a previously unknown set of emergent symbolic mechanisms that support binding specifically in VLMs, via a content-independent, spatial indexing scheme. Moreover, we find that binding errors, when they occur, can be traced directly to failures in these mechanisms. Taken together, these results shed light on the mechanisms that support symbol-like processing in VLMs, and suggest possible avenues for reducing the number of binding failures exhibited by these models.

## 1 INTRODUCTION

The visual world is composed of many commonly recurring elements — shapes, colors, textures, etc. — and visual inputs can be efficiently represented by making use of this compositional structure, for instance, by representing a 'red square' as a combination of features for 'red' and 'square'. There is growing evidence that neural networks, including language models and vision language models (VLMs), learn such compositional representations and use them to represent novel combinations of familiar features (Lepori et al., 2023; Lewis et al., 2022; Campbell et al., 2024; Assouel et al., 2025). This compositional coding approach, however, introduces a fundamental representational challenge, sometimes referred to as the 'binding problem' (Treisman & Gelade, 1980; Greff et al., 2020; Frankland et al., 2021). Given a set of compositional features, such as features representing basic colors and shapes, the binding problem refers to the question of how these features are bound together to represent the entities and relational configurations in a specific context. For instance, in order to represent an image that contains a red square and a blue circle, and to distinguish it from other potential combinations of the same features (e.g., an image containing a blue square and a red circle), it is necessary to form in-context associations between the features that correspond to the same entities (i.e., to bind 'red' to 'square', and 'blue' to 'circle').

Recent work has begun to uncover the representations and mechanisms that support this capacity in language models. Of particular relevance is the discovery of emergent *binding IDs*, vectors representing content-independent indices — akin to symbolic variables — that language models use to track the assignment of entities and attributes in-context (Feng & Steinhardt, 2023; Feng et al., 2024). Binding IDs are additively incorporated into the token embeddings corresponding to their

---

Code will be made available here.

arguments, and can be intervened upon to produce systematic binding errors. Other recent work has identified a set of emergent *symbolic mechanisms* that language models use to perform abstract reasoning (Yang et al., 2025). These mechanisms convert input tokens to abstract, symbol-like variables, which can then be processed independently of the tokens to which they're bound, before eventually being converted back to the corresponding tokens to perform inference. These and other recent findings (Griffiths et al., 2025) suggest that language models rely on an emergent form of symbol processing to represent and reason about entities and relations in-context, but it is an open question whether VLMs employ similar mechanisms to bind *visual* entities in-context.

It is particularly notable that many of the most puzzling shortcomings of VLMs are directly related to the binding problem. VLMs perform very poorly on many tasks that are easy for humans, such as counting or visual search (Rahmanzadehgervi et al., 2024), and many of these tasks involve the need to accurately bind features in-context. Indeed, careful behavioral evaluations have found that VLMs display specific psychophysical signatures similar to those observed in human vision for conditions that interfere with binding (Campbell et al., 2024). These findings underscore the importance of understanding the mechanisms that support binding in VLMs, and especially how the failure of these mechanisms might explain the binding failures that limit the performance of these models.

In this work, we present evidence for a set of emergent symbolic mechanisms that support binding in VLMs. Similar to the content-independent binding IDs that have been identified in text-only language models, we find that VLMs use *visual space* as a content-independent scaffold to bind features and parse multi-object scenes. We therefore refer to these indices as *position IDs*. We also identify a set of *visual symbolic mechanisms* that VLMs use to manipulate these indices: 1) *ID retrieval heads* retrieve the position ID associated with an object described in the prompt, based on the features of that object; 2) *ID selection heads* compute the ID of a target object; and 3) *feature retrieval heads* use this ID to retrieve the features associated with that object. We present convergent evidence for these mechanisms across a set of representational, causal mediation, and intervention analyses. Finally, we analyze the role that these mechanisms play in the persistent binding failures exhibited by VLMs. We find that binding errors can be directly tied to failures of the identified mechanisms. Taken together, these results begin to uncover the mechanisms that support symbol-like binding and inference in VLMs, and illustrate how the breakdown of these mechanisms contributes to their persistent binding failures, suggesting avenues for further improvement of these models. Our specific contributions are as follows:

- We identify and characterize the role of 3 sets of attention heads involved in visual object binding (Section 3). We define the sets using causal mediation analyses (Section 3.4).
- We validate the role of these attention heads and the use of position IDs across a diverse range of VLMs (7 models) through representational analyses(Section 3.3) and intervention experiments (Section 4.2).
- We show the generality and reuse of position IDs across several tasks and in photorealistic images (Sections 4.1, 4.3, and 4.4).
- We link binding failures in VLMs to interference during the ID retrieval process, suggesting new avenues for improving visual grounding in VLMs (Section 4.5).

## 2 RELATED WORK

A number of studies have investigated the emergent mechanisms that support symbol-like processing in language models and other neural networks. This work has identified a number of surprisingly interpretable and structured mechanisms, including induction heads (Olsson et al., 2022), function vectors (Todd et al., 2023), binding IDs (Feng & Steinhardt, 2023; Feng et al., 2024), and emergent symbolic mechanisms (Yang et al., 2025). Related work has found convergent evidence for such emergent mechanisms in transformer language models and vision transformers that are trained in controlled settings (Lepori et al., 2024; Tang et al., 2025). Although a number of recent studies have begun to apply mechanistic interpretability techniques to understand VLMs (Neo et al., 2024; Golovanevsky et al., 2024; Kaduri et al., 2024; Basu et al., 2024), it has not yet been established whether VLMs possess emergent mechanisms for symbol processing similar to those that have been identified in text-only language models, as our results suggest.

Beyond the difference in modalities (images vs. text), there are also several novel contributions of our work that go beyond the previously identified emergent symbolic mechanisms in text-only

language models (Yang et al., 2025). First, although the identified mechanisms in both cases involve an emergent form of symbol processing, the specific function that these mechanisms perform is different (parsing of multi-object scenes vs. induction of abstract relational patterns). This is not merely a translation of the same circuit from the textual domain into the visual domain, but entails a different circuit altogether. Second, unlike previous results in text-only models, the emergent visual symbolic representations are distributed across multiple tokens (each object spans multiple patches), demonstrating that emergent symbol processing extends to more naturalistic and high-dimensional domains. Third, the identified visual symbolic mechanisms are directly related to a significant limitation faced by current VLMs (the binding problem), suggesting that these mechanisms may be of practical relevance for improving these models.

Several studies have probed the capacity of VLMs to process multi-object scenes, revealing a number of failure modes for tasks such as counting (Rahmanzadehgervi et al., 2024; Rane et al., 2024; Zhang & Wang, 2024), visual search (Campbell et al., 2024), and visual analogy (Mitchell et al., 2023; Yiu et al., 2024; Fu et al., 2025). Notably, all of these failure modes appear to be directly related to difficulty with binding object features and relations (Yuksekgonul et al., 2023; Campbell et al., 2024; Assouel et al., 2025). Our results reveal some of the key mechanisms that are involved in these binding failures, suggesting potential avenues for further improvement of VLM architectures and training.

Finally, our findings are closely related to work that has identified the neural and psychological correlates of visual binding. In cognitive science, visual indexing theory (Pylyshyn, 2001) has postulated the existence of content-independent, visual indices that are used for binding object features. In neuroscience, there is a broad distinction between brain regions involved in representing concrete features (e.g., shapes and colors), and brain regions involved in representing space (Goodale & Milner, 1992). The spatial representations in this latter set of brain regions also appear to be more broadly involved in abstract, symbol-like processing (Whittington et al., 2020; O'Reilly et al., 2022; Webb et al., 2024). The emergent mechanisms that we have identified in VLMs, in which visual space serves as a content-independent scaffold for binding object features, thus have interesting parallels to findings from cognitive science and neuroscience.

# 3 SYMBOLIC MECHANISMS FOR VISUAL BINDING IN VLMS

## 3.1 SCENE DESCRIPTION TASK

To identify the mechanisms underlying visual binding in VLMs, we used a scene description task that tests the model's ability to bind features to objects in multi-object visual scenes. In that task, the model receives an image containing multiple objects (e.g., colored shapes) alongside a prompt describing some, but not all, objects present. The model must identify and describe the missing object. For instance, given an image with a red square, blue circle, and green triangle, plus the prompt *"This image contains a red square, a blue circle and a "*, the model must respond *"green triangle."*

This task requires the model to: (1) parse the visual scene, (2) match objects described in the prompt to visual representations, (3) determine the missing object, and (4) retrieve its features. Crucially, success in this task demands accurate spatial binding to avoid erroneously combining features across objects.

## 3.2 POSITION ID ARCHITECTURE

We identified a three-stage architecture that VLMs employ to solve the binding problem (Figure 1a). These stages implement a content-independent indexing scheme using **position IDs**—spatial indices serving as symbolic variables for tracking visual objects.

**Stage 1: Position ID Retrieval** ■ In this stage the model establishes correspondences between semantic content in the prompt and spatial locations in the image. Given an object description (e.g., "red square"), these heads retrieve the position ID for that object from the corresponding image tokens. The output of these heads is an abstract spatial index—not the object's features, but a *pointer* to its location.

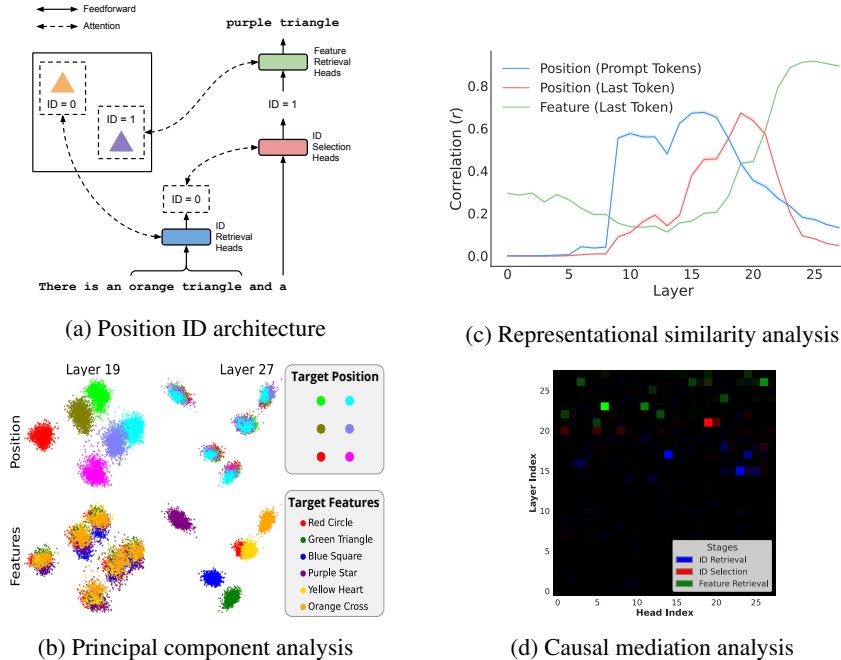

(a) Position ID architecture

(c) Representational similarity analysis

(b) Principal component analysis

(d) Causal mediation analysis

Figure 1: Overview of position ID architecture and supporting evidence. We identify three processing stages: ID retrieval heads that retrieve the position ID of objects described in the prompt, ID selection heads that select the position ID for a target object, and feature retrieval heads that use this position ID to retrieve the features of the target object. These stages are revealed both by (b,c) representational and (d) causal mediation analyses (see text for explanation).

**Stage 2: Position ID Selection** ■ In this stage, the model selects the position ID for the target object, i.e., the object that will be described next, based on the position IDs that have already been retrieved in the preceding prompt.

**Stage 3: Feature Retrieval** ■ In this stage, the model uses the position ID from stage 2 as an index to retrieve semantic features of the target object from the image tokens corresponding to that object.

In the following sections, we first present representational analyses (Section 3.3) that confirm the progression of these three stages across layers and sequence positions, and then describe the causal mediation procedure (Section 3.4) used to identify the specific attention heads that implement these stages. We show results for Qwen2-VL in the main text, but results for additional models are shown in the Appendix sections A.5.1 and A.5.2).

## 3.3 REPRESENTATIONAL ANALYSES

We carried out representational analyses using two approaches: 1) principal component analysis (PCA), and 2) representational similarity analysis (RSA). The PCA results for Qwen2-VL are shown in Figure 1b. These results provide an intuitive visualization of stages 2 and 3. The results show the hidden state activations for the last token in the sequence, at which the model must generate a description of the target object. These activations are projected on to the top 2 principal components (PCs), and colored according to either the spatial position of the object (top row), or the features (color and shape) of the object (bottom row). The results show that spatial position is clearly separable in the embeddings for layer 19 (left column), whereas object features are completely overlapping. This corresponds to stage 2, during which the model computes the position ID for the target object. By contrast, in layer 27 (right column), spatial position is overlapping, while object features become separable. This corresponds to stage 3, during which the model retrieves the features of the target object based on its ID.

To more comprehensively characterize this process, we also performed RSA Kriegeskorte et al. (2008). In RSA, two embedding spaces can be compared by first computing the pattern of pairwise similarities within each space, and then computing the correlation between these similarities. Using this approach, we compared the hidden state embeddings in the model to two hypothesized embedding spaces, one that coded only object position, and one that coded only object features. The results (Figure 1c) further confirmed the proposed three-stage progression (Figure 1a). At the prompt tokens describing an object, the position of that object was most strongly represented in layers 14-17, as predicted by the ID retrieval stage. At the final token, the position of the target object was most strongly represented in layers 18-21, as predicted by the ID selection stage, and the features of the target object were most strongly represented in layers 23-26, as predicted by the feature retrieval stage. We also found similar RSA results for the other models studied (Appendix A.5.1)

## 3.4 Identifying Attention Heads via Causal Mediation Analysis

In the previous section, we presented representational evidence consistent with the proposed three-stage architecture. To identify the specific mechanisms that implement these stages, we next performed causal mediation analysis (CMA) (Pearl, 2022; Wang et al., 2022; Meng et al., 2022; Yang et al., 2025). This approach allows us to estimate the causal effect of an embedding at a particular layer, position, or attention head. In our case, given our hypothesized architecture, we focus on analyzing the *causal effect of the output of attention heads* in the prompt tokens. Our goal in this analysis was to identify the attention heads that are causally involved in the 3 stages that we identified: 1) retrieving the ID of the object mentioned in the prompt, 2) computing the ID of the missing target object, and 3) using this ID to retrieve the features of the target object. We refer to the heads that perform these computations as **ID retrieval heads**, **ID selection heads**, and **feature retrieval heads** respectively.

In order to isolate these sets of heads, we designed three conditions. Each CMA condition is defined by a clean context $c_1$, with correct answer $a_1$, and a modified context $c_2$. Activations are patched from $c_2$ to $c_1$, yielding the patched context $c_1^*$, with expected answer $a_1^*$. Given a model $M$, we measure the causal mediation score as defined by (Wang et al., 2022):

$$ s = (M(c_1^*)[a_1^*] - M(c_1^*)[a_1]) - (M(c_1)[a_1^*] - M(c_1)[a_1]) \tag{1} $$

where $M(c)[a]$ corresponds to the logits of token $a$ at the output of the model evaluated on input $c$. Intuitively, this score measures the extent to which patching activations from $c_2$ to $c_1$ has the expected effects on the model's outputs (i.e., makes the model more likely to respond with $a_1^*$ than $a_1$). We performed causal mediation on a simpler version of the scene description task involving only two objects. All CMA scores are averaged across 50 samples for each condition. Below we define the three different conditions corresponding to the three hypothesized stages. We include a schematic illustration of the CMA patching intervention targeting all sets of head in Appendix A.3.

**ID Retrieval Heads**  To identify heads responsible for retrieving the position ID of objects mentioned in the prompt, we designed a condition where the clean context $c_1$ contains an image with two objects and a prompt mentioning one of them, with the clean answer $a_1$ being the color of the unmentioned object. The modified context $c_2$ uses the same prompt but with the object positions swapped in the image. We predicted that this would result in the position IDs assigned to these objects also being swapped. We performed causal mediation by patching attention head outputs at the prompt tokens for the object described in the prompt. If an attention head is causally involved in retrieving the position ID for the described object, then patching from the modified context (where object positions are swapped) should cause the model to retrieve the wrong position ID (namely, the position ID for the target object, rather than the object described in the prompt). This in turn should cause the ID selection heads to erroneously select the position ID for the object already described in the prompt (rather than the target object), which should ultimately cause the feature retrieval heads to retrieve the features for the object already described in the prompt (see Figure 10 for a schematic depiction). The expected answer $a_1^*$ in this condition is therefore that the model should repeat the features for the object already described in the prompt.

**ID Selection Heads**  To identify heads that compute the position ID of the target object, we used the same clean and modified contexts as in the ID retrieval condition. However, we performed

causal mediation by patching attention head outputs at the *last token position*, where the model must generate a description of the target object. If an attention head is causally involved in computing the target object's position ID, then patching from the modified context (where object positions are swapped) should cause these heads to select the wrong target ID (namely, the ID for the object already mentioned in the prompt). This in turn should cause the feature retrieval heads to retrieve the features associated with the object already described in the prompt (see Figure 11), and therefore the expected answer $a_1^*$ in this condition is that the model should repeat the features for the object already described in the prompt.

**Feature Retrieval Heads**   To identify heads that retrieve object features, we used clean and modified contexts that differed only in the features of the target object. The features and position of the object mentioned in the prompt, as well as the position of the target object, were the same for both contexts. We patched attention head outputs at the last token position. If an attention head is causally involved in retrieving the features of the target object, then patching from the modified context (in which the target object has different features) should cause the model to retrieve the wrong feature information (see Figure 12), and the expected answer $a_1^*$ is therefore the semantic features of the target object in the modified context rather than the original target object.

**CMA Results**   We performed CMA for these 3 conditions on all models (Qwen2.5-VL-3B, Qwen2.5-VL-7B, Qwen2.5-VL-32B, Qwen2-VL, Llava1.5-7B, Llava1.5-13B, and Llava-OneVision-7B) and report the results in the Appendix (see Figures 20-25) . We include the CMA results for Qwen2-VL in Figure 1d, where the CMA scores are color coded by condition (i.e., head type). These results confirmed the predicted three-stage progression, with ID retrieval heads (in blue) primarily localized to layers 12-16, ID selection heads (in red) primarily localized to layers 18-19, and feature retrieval heads (in green) primarily localized to layers 20-27. These layers also corresponded closely to the layers identified for these stages in the representational analyses.

## 4   RESULTS

Having identified the mechanisms that support binding in VLMs, we next performed a series of analyses to characterize these mechanisms in more detail. These analyses provide a richer picture of the coding scheme that they employ, demonstrate their generalizability across tasks and image types, and implicate them in the binding failures displayed by VLMs.

### 4.1   ARE POSITION IDS RELATIVE OR ABSOLUTE?

In this section we focus on determining whether position IDs employ a *relative* or *absolute* spatial coding scheme. We designed an experiment using a 3×3 object grid. We created four separate 2×2 configurations, each positioned within different quadrants of the larger (3x3) grid (Figure 2). Crucially, across all four arrangements, one object was always placed at the same absolute grid location (the center position, circled in Figure 2). We prompted the model to perform the scene description task from the previous analyses, in which a single object is missing from the prompt. We analyzed the representations at the last token, where the model must predict the missing object.

We then performed RSA using similarity matrices that reflected either relative (within each inner 2x2 grid) or absolute position (within the larger 3x3 grid). We found that models from the Qwen family had a clear preference for a relative coding scheme (Figure 2 shows results for Qwen2-VL). The effects were less pronounced for models from the LLava family (see Appendix A.5.1 for results from all models). This may be a result of differences in position embeddings for the Llava models, or may be due to Llava's significantly smaller training set. Finally, we further confirmed these results with a set of interventions that targeted relative vs. absolute positions (see Appendix A.1.2, Figures 13 and 30).

### 4.2   POSITION IDS GENERALIZE TO VISUALLY COMPLEX SETTINGS

We next investigated whether the symbolic mechanisms employed in synthetic visual tasks are also used when processing more complex, naturalistic images. To do so, we generated a dataset using the Photorealistic Unreal Graphics (PUG (Bordes et al., 2023)) environment, which allows for

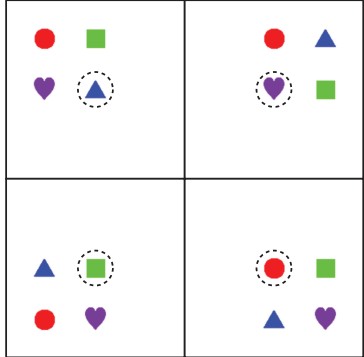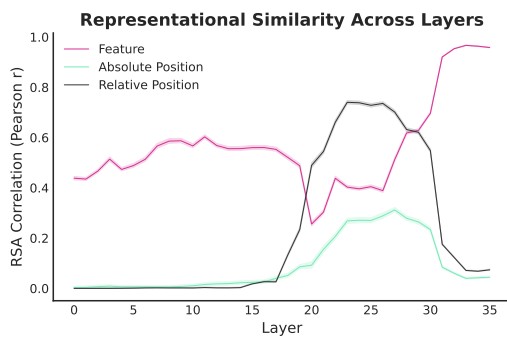

Figure 2: Testing for relative vs. absolute spatial coding. Left: Example of each grid condition with centrally located object circled. Right: RSA results showing second-order correlation of last token embeddings (Qwen2-VL) with relative position, absolute position, and object features.

the generation of images that capture important properties of real-world images, including three-dimensional structure, occlusion, and lighting/shadows. Our dataset was comprised of images that each contained two distinctly colored 3D animals in varying realistic backgrounds (see Appendix A for details). We randomly jittered the positions of the animals, but ensured that each image has a clear leftmost and rightmost animal (to ensure that we have ground-truth labels for the position IDs that the models will assign).

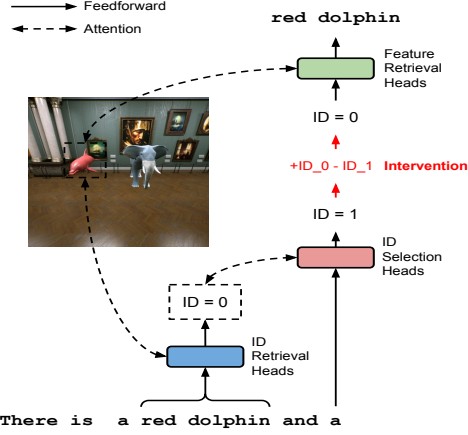

(a) Intervention targeting ID selection heads.

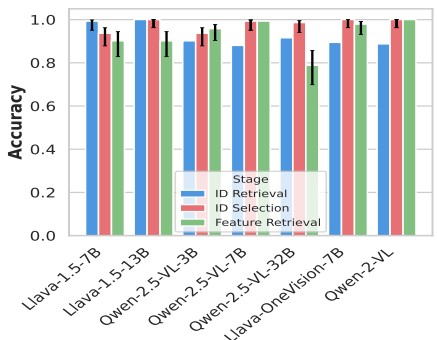

(b) Intervention accuracy across all models and sets of heads.

Figure 3: Photorealistic intervention results targeting the three position ID processing stages as identified by CMA.

Our position ID hypothesis predicts that by intervening on the position IDs, we can steer the model to predict the color of either of the animals based solely on position ID (left vs. right) assigned to them. Figure 3a shows an illustration of this intervention for the outputs of the ID selection heads. The basic logic of the intervention is to first estimate the embeddings for position IDs 0 and 1 (by averaging over several instances of these IDs), and then to 'edit' the IDs computed by the model (i.e., to subtract the ID computed by the model and add the other ID).

We performed this intervention targeted to each of the 3 identified sets of heads. For the ID retrieval and selection heads, we performed the intervention on the output of the attention heads. For the feature retrieval heads, we performed the intervention on the queries, since our hypothesis is that these heads use position IDs as queries to retrieve features stored in the image tokens.

The intervention can be formally defined as follows:

$$\tilde{o}_h(x) = o_h(x) + \alpha * (d_t - d_o)$$

$$d_{i \in t, o} = \mathbb{E}_{x \in X_i}[o_h(x)]$$

where $o_h(x)$ is the output of attention head $h$ given input $x$, $d_t$ is the estimate of the target binding ID (e.g., ID 0 in Figure 3a), $d_o$ is the estimate of the original binding ID (e.g., ID 1 in Figure 3a), and $\alpha$ is a coefficient that controls the magnitude of the intervention. The intervention was applied to the top $K$ heads as defined by the CMA conditions for each head type. We optimized the intervention by sweeping over a range of values for $\alpha$ and $K$ (see Appendix A.5.5 for results across all values), and show the results for the best performing parameters in Figure 3b. The results indicated that this intervention was highly effective ($> 79\%$ efficacy) for all 3 head types and all models, demonstrating that position IDs also play a central role in processing more complex images. Furthermore, we present intervention results in section B of the Appendix showing that position IDs also play a role in processing of real-world images (using the COCO dataset).

## 4.3 POSITION IDS ARE LOCALIZED WITHIN VISUAL OBJECT PATCHES

We then sought to determine whether position IDs are **localized** in the residual stream of visual patches spanning the objects in the image. This analysis employed a simple color retrieval task, in which the model is prompted to identify the color of a specified object. The following prompt template was used: "In this image what is the color of the {SHAPE}. Answer with the correct color only." Our objective was to steer the models to retrieve the color of an arbitrarily specified object by intervening on the keys of the visual object patches. Let $K_l^o \in R^{N_o \times d}$ be keys vectors spanning the $N_o$ visual patches of object $o$. We perform an additive intervention on the $K_l^{o_0}$ in order to swap the ID of object $o_0$ with the ID of object $o_1$ at layer $l$ such that :

$$\tilde{K}_l^{o_0} = K_l^{o_0} + \alpha(d_1^l - d_0^l)$$

$$\tilde{K}_l^{o_1} = K_l^{o_1} + \alpha(d_0^l - d_1^l)$$

with $d_i = \mathbb{E}_{o \in O_i}[K_l^o]$ and $O_i$ being the set of object patches with ID $i$. Importantly, we performed the intervention *before* the RoPE embedding module, and therefore any effects of this intervention on position IDs is independent of the RoPE position embeddings. For this experiment, we fixed $\alpha$ to 2, and we applied the intervention to the layers containing the top-20 highest scoring feature retrieval heads (according to the CMA scores).

The results are shown in Figure 4. This intervention was highly effective for all models, confirming that position IDs are stored locally in the keys of the image patches spanning each object.

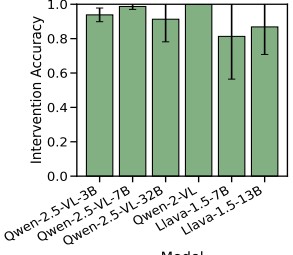

## 4.4 POSITION IDS ARE REUSED ACROSS TASKS

We next sought to determine whether position IDs are also used in more complex tasks. To do so, we focused on a visual reasoning task involving spatial relations. The task involved questions with the following template "In this image, what is the color of the object that is directly {RELATION} of {REF}. Answer with the relevant color

Figure 4: Color Retrieval Intervention Accuracy for all models.

only." where RELATION was one of {above, below, left, right} and REF was one of the reference objects in the image. Importantly, we first estimated position IDs based on the *scene description* task, and then applied these IDs in an intervention in the visual reasoning task, thus assessing the extent to which *position IDs are reused across tasks*. More precisely, given the output $o_h(x)$ of head $h$ with input context $x$, we performed an additive intervention such that

$$\tilde{o}_h(x) = o_h(x) + \alpha * d_{\text{desc}}^i$$

where $d_{\text{desc}}^i = \mathbb{E}_{x \in X_i}[o^h(x)]$, with $X_i$ being the set of scene description input context in which the target object is assigned to ID $i$. The intervention was applied to the top 100 ID selection heads (based on the CMA scores), and $\alpha$ was set by 2.

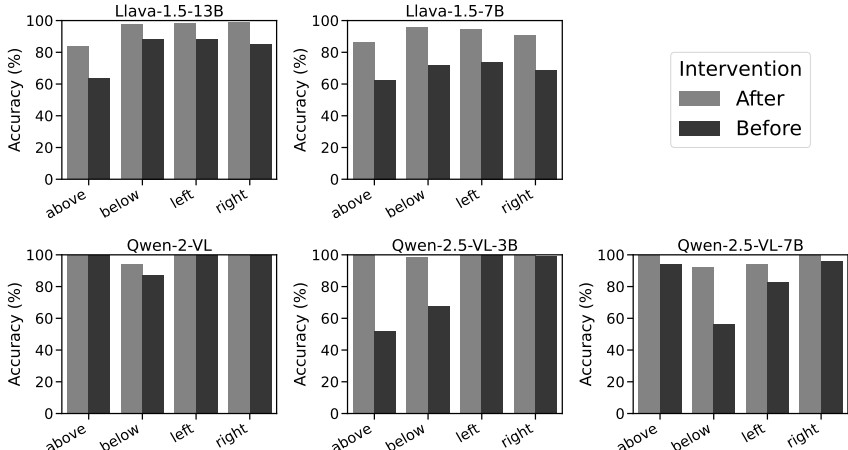

Figure 5: Intervention performance on spatial reasoning task.

Figure 5 shows the effects of this intervention for 5 out of 7 models (Qwen2.5-32B and Llava-OneVision were already at ceiling-level performance on this task, making an intervention uninformative). The results show that transferring position IDs from the scene description task – where models generally performed well – significantly improved performance on the more challenging spatial reasoning task. These results highlight not only that position IDs are employed in more complex tasks, such as spatial reasoning, but that the *same* position IDs are used across tasks, indicating a general-purpose symbolic indexing scheme.

## 4.5 BINDING ERROR ANALYSIS

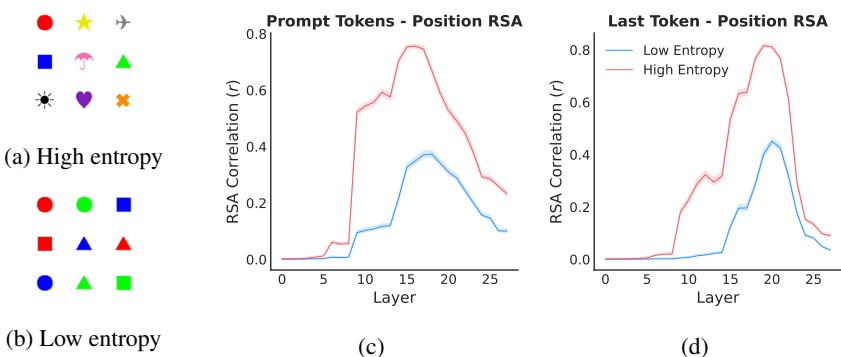

Figure 6: Binding error analysis showing effect of feature entropy on position ID mechanisms.

We next investigated the role that position IDs and visual symbolic mechanisms play in binding errors. Specifically, we analyzed how position IDs influence the model's ability to bind features to objects or locations, using a scene description task with varying feature entropy – a factor known to impact binding errors (Greff et al., 2020; Campbell et al., 2024). Details of the generation process are provided in Appendix A.5.4 and examples of low (resp. high ) entropy stimuli are shown in Figure 6b (resp. 6a). Our results (Figure 6) show that entropy level strongly affects position ID representations at two critical stages: ID retrieval Figure 6c) and ID selection (Figure 6d). We observe that lower entropy leads to less accurate ID retrieval and more ambiguous ID selection, reflected in performance differences between low and high entropy settings (Figure 7). We also report overall RSA (averaged across layers) for prompt tokens and last tokens in Figure 7. Layer-wise RSA details for all models are included in Appendix A.5.4. We also show the causal relation of the binding ID mechanisms failure and the binding errors in this task in Appendix D

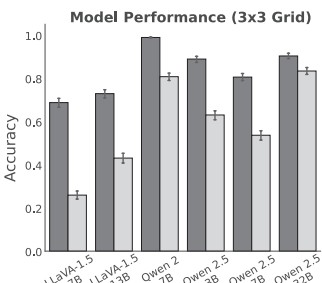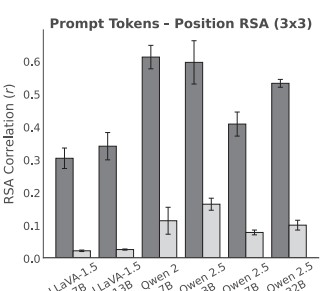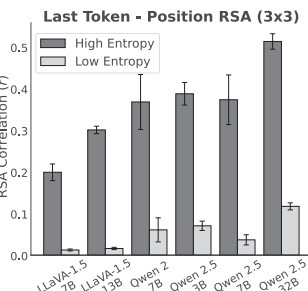

Figure 7: Link between binding errors and failure of position ID mechanisms.

## 5 CONCLUSION AND FUTURE WORK

In this work, we have identified and characterized a set of emergent symbolic mechanisms that VLMs use to perform visual binding (Greff et al., 2020; Lewis et al., 2022; Campbell et al., 2024; Assouel et al., 2025) in multi-object scenes. Through a combination of representational analyses, causal mediation, and interventions across seven different VLM models, we uncovered a three-stage architecture that employs **position IDs** as content-independent spatial indices for binding object features. These mechanisms are similar to mechanisms that support symbol-like processes in language models, most notably including binding IDs (Feng & Steinhardt, 2023) and emergent symbolic mechanisms (Yang et al., 2025), but our results extend these findings to visual processing.

Our key findings demonstrate that VLMs implement binding through: (1) **ID retrieval heads** that establish correspondences between semantic content in prompts and spatial locations in images, (2) **ID selection heads** that compute the position ID of target objects, and (3) **feature retrieval heads** that use these IDs to retrieve object features. Importantly, we show that this architecture is remarkably consistent across multiple model families and scales, suggesting it represents a fundamental solution to visual binding in current VLMs. Our analyses reveal several critical properties of these mechanisms: position IDs employ relative rather than absolute spatial coding; they generalize beyond synthetic stimuli to complex, photorealistic images with naturalistic backgrounds; they are localized within the image patches corresponding to each object; and they are reusable across different tasks, indicating a *general-purpose symbolic indexing system* that supports diverse forms of visual reasoning.

Crucially, we demonstrate that the persistent binding failures exhibited by VLMs can be directly traced to failures in these symbolic mechanisms. We find that position IDs are less accurately represented in conditions that typically lead to binding errors, such as images where multiple objects share features. These findings have important implications for the development of more capable VLMs. Our results suggest that binding performance may be improved either via architectural innovations that better support spatial indexing (such as object-centric architectures (Locatello et al., 2020)) or training strategies that explicitly strengthen these symbolic mechanisms, such as spatial pointing tasks (Deitke et al., 2024).

It is worth considering the extent to which the identified mechanisms are truly emergent, particularly given the potential relationship between a model's innate position embeddings and the structure of the position IDs. It is important to emphasize that these mechanisms involve several operations that are not entailed by position embeddings alone, and which are not built into the architecture, including the clustering of position embeddings for patches belonging to the same object, and the specific function of the three identified attention heads (mediated by the content of their queries, keys, and values). However, it is an open question whether the emergence of these mechanisms may be driven by architectural inductive biases, such as the use of distinct query/key and value embeddings (enabling a form of indirection, or the use of 'pointers'), or distributional aspects of the training data. We leave the investigation of these questions to future work.

## 6 REPRODUCIBILITY STATEMENT

We included all the generation details of our synthetic datasets in Appendix A. A detailed explanation of the RSA is provided in A.2 and the CMA score in Section A.5.2 of the main text. We additionally referenced all the analyses for all the models we study in Appendices A.5.5, A.5.2, and A.2. Datasets, analysis and intervention code will be released.

## 7 ACKNOWLEDGMENTS

RA would like to thank Sjoerd Van Steenkiste for all the brainstorming sessions during the course of the project. DC is supported by the NSF GRFP Fellowship and the Natural and Artificial Minds Graduate Fellowship. TW is supported by the Courtois Chair in Fundamental Research V (Neuroscience).

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

# A  APPENDIX

## A.1  DATASET GENERATION DETAILS

### A.1.1  SYNTHETIC SCENE DESCRIPTION TASK

This section describes the synthetic image datasets generated for representational similarity analysis (RSA), patching experiments, and principal component analysis (PCA). All images used colored shapes that each occupied a 56×56 pixel region, comprising a 2x2 grid of patches (each patch had a size of 28x28 pixels). RSA and intervention experiments were conducted using datasets generated by arranging objects within a 2x2 or 3x3 grid, while PCA analyses used a 3×2 grid configuration.

**Dataset generation protocol:**  For each dataset, there were N unique objects defined by unique color-shape conjunctions, where N was equal to the number of grid positions. Each image contained one target object at a fixed position, with the remaining N-1 objects randomly permuted across remaining positions. We generate K trials for each combination of object identity and grid position, yielding K × N × N total trials per dataset.

**PCA dataset**  The 3×2 grid configuration (392×932 pixels) used for the PCA results shown in Figure 1b used 6 specific color-shape conjunctions: red circle, green triangle, blue square, purple star, yellow heart, and orange cross. Objects were placed in the leftmost and rightmost columns of a 3×3 grid layout with the center column empty. 200 trials/combination were generated, resulting in 7,200 total trials.

### A.1.2  DATASET FOR TESTING ABSOLUTE VS. RELATIVE SPATIAL CODING

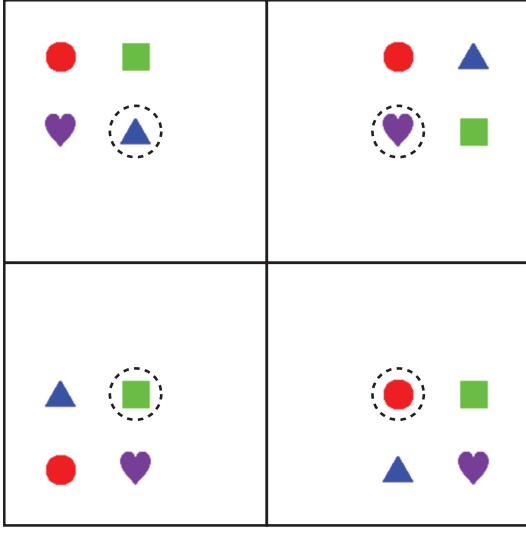

Figure 8: Stimulus condition for characterizing relative vs absolute position IDs.

To determine whether position IDs use relative or absolute position coding, we designed an experiment using a 3×3 object grid. We created four different 2×2 sub-arrangements of objects, each positioned within different quadrants of the larger grid (Figure 8). Crucially, across all four arrangements, one object was always placed at the same absolute position (the center position).

### A.1.3  SCENE DESCRIPTION TASK WITH NATURALISTIC IMAGES (PUG)

To validate the use of position IDs in more naturalistic settings, we performed additional experiments using the PUG (Photorealistic Unreal Graphics) environment, which allows for the generation of images that capture important properties of real-world images, including three-dimensional structure, occlusion, and lighting/shadows. We generated a dataset of images that each contained two distinctly colored 3D animals with realistic backgrounds. We randomly jittered the positions of the animals, but ensured that each image has a clear leftmost and rightmost animal (to ensure

that we have ground-truth labels for the position IDs that the models will assign). An example is shown in Figure 3. The dataset is comprised of 200 images with 3 different animals (camel, dolphin, elephant) and 3 different colors (green, red, white) in 3 different realistic environments (beach, salt desert, museum).

### A.1.4 DATASET FOR INVESTIGATING BINDING ERRORS

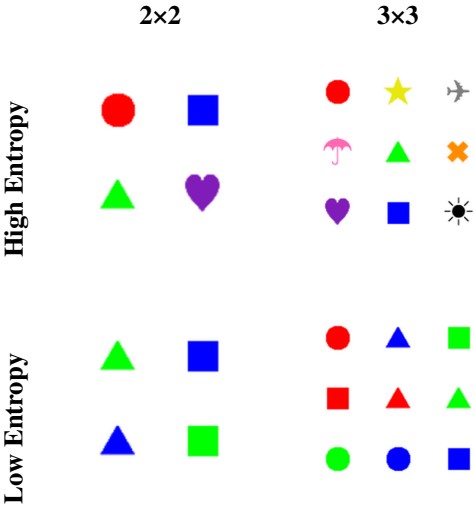

Figure 9: Image configurations for investigating binding errors.

To investigate the role of position IDs in binding errors, we created datasets varying in grid size and entropy. Low entropy datasets used all possible conjunctions from limited color and shape sets, while high entropy datasets used the most distinct conjunctions from larger sets. The 2×2 grid (280×280 pixels; 4 objects) used 2 colors × 2 shapes for low entropy and the 4 most distinct conjunctions from 4 colors × 4 shapes for high entropy (both involving 100 trials/combination; 1600 total). The 3×3 grid (392×392 pixels; 9 objects) used 9 conjunctions from 3 colors × 3 shapes for low entropy and 9 objects with unique color and shape for high entropy (both involving 50 trials/combination; 4,050 per condition).

## A.2 REPRESENTATIONAL SIMILARITY ANALYSIS PROTOCOL

We used representational similarity analysis (RSA) to assess the alignment of the model's internal representations with either the spatial position or the semantic features (color and shape) of objects. RSA quantifies representational alignment by comparing pairwise representational similarity matrices (RSMs) between model embeddings and hypothesized representational structures, as quantified by correlation metrics.

### A.2.1 TOKEN POSITIONS USED FOR RSA

We performed RSA on embeddings from two different token position sources. The first was the last token in the sequence, positioned immediately before the description of the target object. This token represents a critical decision point where the model must integrate all relevant information to generate the appropriate object description. Analyzing this single token position yields RSMs of shape [1, T, T], where T represents the number of trials. The second source was the comma tokens that punctuate each object description in the prompt. Specifically, we analyze the comma tokens that follow each of the N-1 object descriptions preceding the target. These tokens serve as natural boundaries between object descriptions and we found that they encode information about the preceding object in the description. This multi-token approach produces RSMs of shape [N-1, T, T]. For both token position sources, we extract activations from two locations within the model: the residual stream and the attention block outputs.

### A.2.2 MODEL RSM CONSTRUCTION

We constructed model RSMs by computing pairwise cosine similarities between activations across samples. All our RSAs were performed on the residula stream. We systematically extract activations at the specified token positions, yielding a single representation per layer. Given a set of activations $\text{Act}(i, t)$ for token $i$ and trial $t$, we computed pairwise similarities across all trial pairs. Each entry in the resulting RSM reflects the similarity of representations across two trials.

### A.2.3 TARGET RSM CONSTRUCTION

To evaluate what type of information these representations encode, we constructed two types of target RSMs. The position-based RSM captures spatial relationships using ground-truth $(x, y)$ coordinates $\text{coord}(i, t)$ of object $i$ in trial $t$, with normalized Euclidean distances:

$$\text{RSMpos}[i, t_1, t_2] = 1 - \frac{D(\text{coord}(i, t_1), \text{coord}(i, t_2))}{\max t_1, t_2 D} \tag{2}$$

The feature-based RSM represents semantic similarity through visual attributes. We constructed separate matrices for color and shape attributes, then combined them:

$$\text{RSMcolor}[i, t_1, t_2] = \mathbb{1}(\text{color}(i, t_1) = \text{color}(i, t_2)) \tag{3}$$

$$\text{RSMshape}[i, t_1, t_2] = \mathbb{1}(\text{shape}(i, t_1) = \text{shape}(i, t_2)) \tag{4}$$

$$\text{RSMfeat}[i, t_1, t_2] = \frac{1}{2}(\text{RSMcolor}[i, t_1, t_2] + \text{RSM}_{\text{shape}}[i, t_1, t_2]) \tag{5}$$

### A.2.4 QUANTIFYING ALIGNMENT

We quantified the alignment between model representations and object features by computing Pearson correlations between model RSMs and each target RSM. These correlations produce scalar similarity scores that vary depending on the source of the activations — for residual stream analysis, we obtained one score per layer, while for attention block analysis, we obtained one score per head per layer. By comparing these alignment patterns across our two token sources, and between the residual stream and attention mechanisms, we can trace how different types of object information are encoded and transformed throughout the model's processing hierarchy.

## A.3   ILLUSTRATION OF CMA CONDITIONS

For the CMA analysis all of the prompts follow the simple Scene description template `In this image there is a [COLOR][SHAPE] and a` and the model is tasked to complete the sentence with the missing colored shape.

Here we present schematic illustrations of the different CMA conditions used to identify the different sets of attention heads described in Section 3.4, namely the ID retrieval heads (Figure 10), ID selection heads (Figure 11), and the feature retrieval heads (Figure 12).

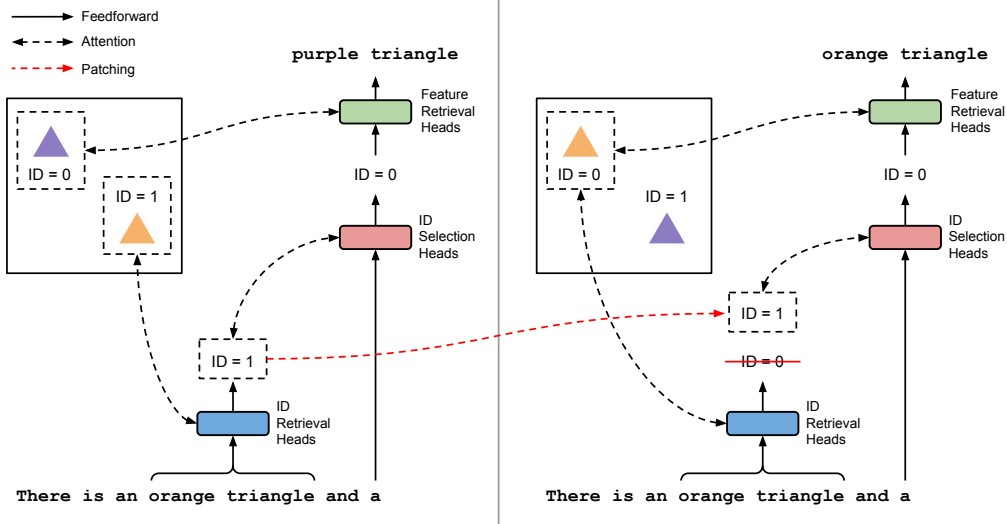

Figure 10: Causal mediation procedure used to identify ID retrieval heads.

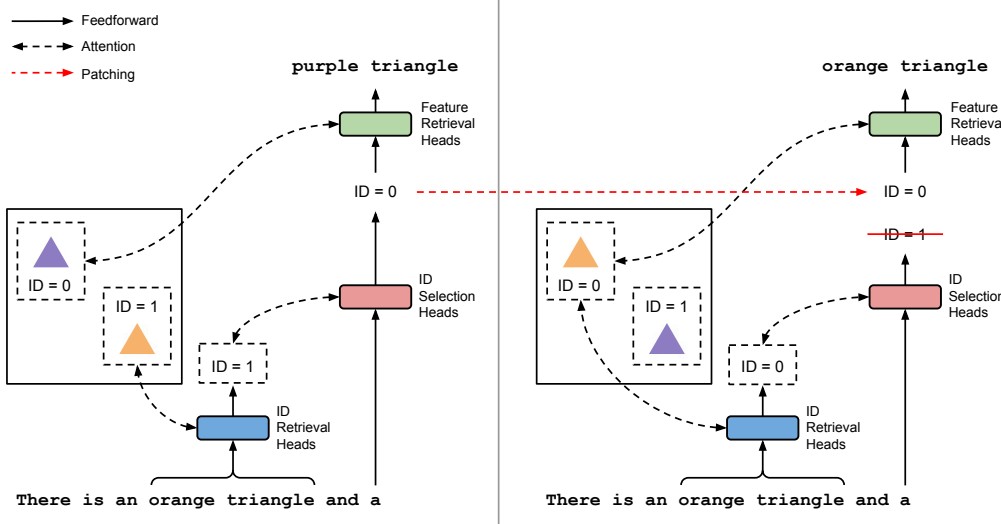

Figure 11: Causal mediation procedure used to identify ID selection heads.

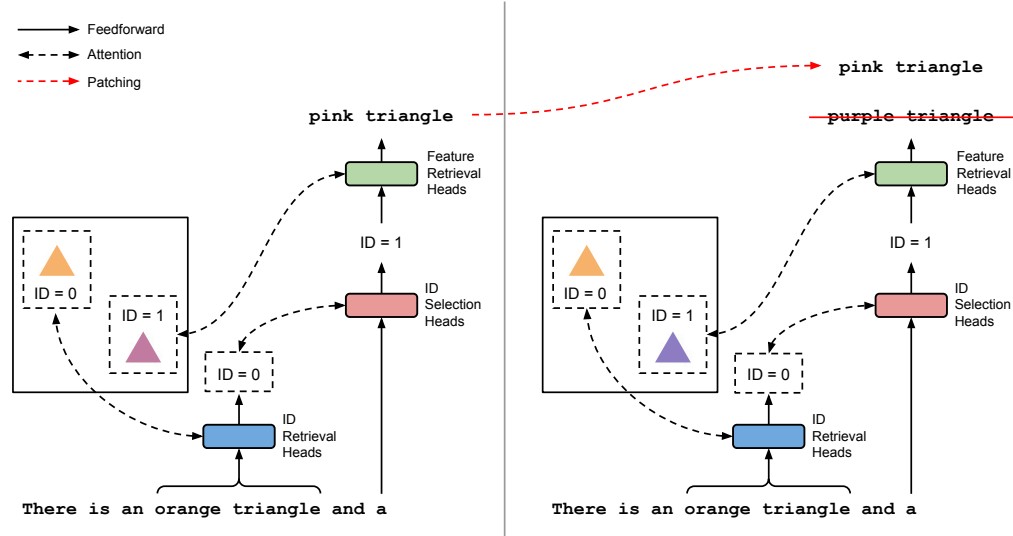

Figure 12: Causal mediation procedure used to identify feature retrieval heads.

## A.4 ILLUSTRATION OF ABSOLUTE VS. RELATIVE POSITION INTERVENTION

Figure 13 illustrates the intervention performed to test whether position IDs employ an absolute or relative spatial coding scheme. The figure depicts a pair of images involving two different 2x2 grid configurations. For the first image, a prompt was presented including 3 out of the 4 objects present in the image (specific prompt: In this image there is a [COLOR0][SHAPE0], a [COLOR1][SHAPE1], a [COLOR2][SHAPE2] and a), and the output of the ID selection heads was extracted at the final token position (where the model generated a description of the target object). For the second image, a prompt was presented including 2 out of the 4 objects (specific prompt: In this image there is a [COLOR0][SHAPE0], a [COLOR1][SHAPE1], a). One of the missing objects appeared at the same *absolute* position as the target object in the first image (the center of the image), and the other missing object appeared at the same *relative* position as the target object from the first image (the lower right quadrant of the 2x2 configuration). The ID for the target object in the first image was patched into the output of the ID selection heads for the target object in the second image, and we assessed whether the object generated by the model matched the prediction of the absolute vs. relative position hypotheses. The results are shown in Figure 30

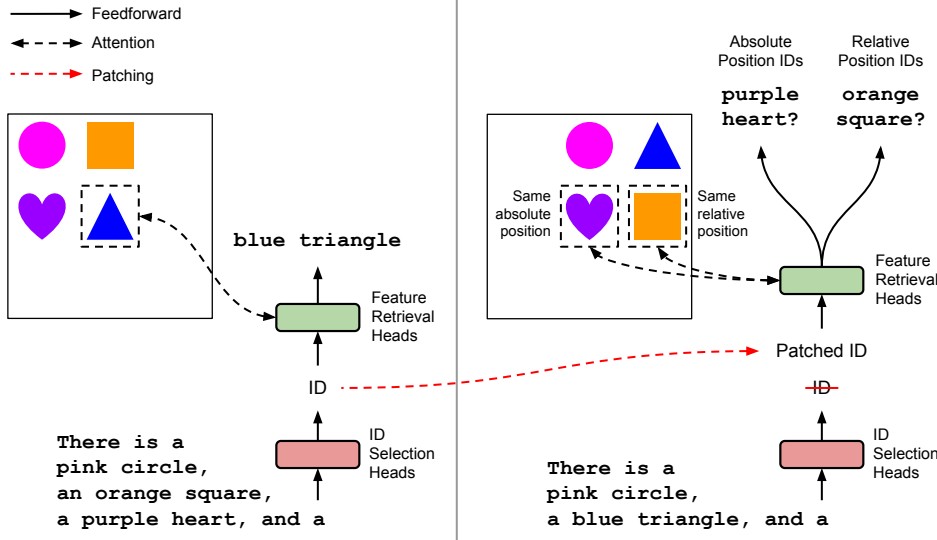

Figure 13: Illustration of the intervention testing for relative vs. absolute spatial coding.

## A.5 ADDITIONAL RESULTS

### A.5.1 REPRESENTATIONAL ANALYSES

Here we present RSA results across a range of different open-source VLMs (Qwen2-vl, LLaVa 1.5, LLaVa-Onevision-7b, and Qwen 2.5) and model scales (Qwen 2.5 3b, 7b, 32b). We find convergent evidence for the same two stage processing (position followed by object features) across all models and model scales.

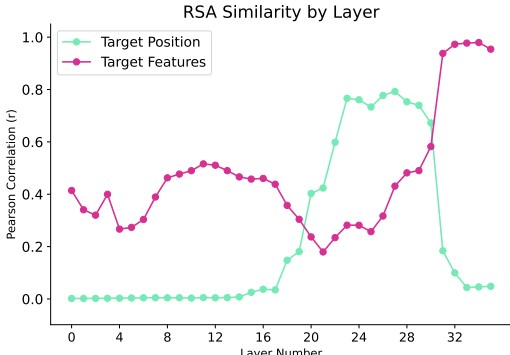

Figure 14: Representational analyses for Qwen2.5-VL-3b.

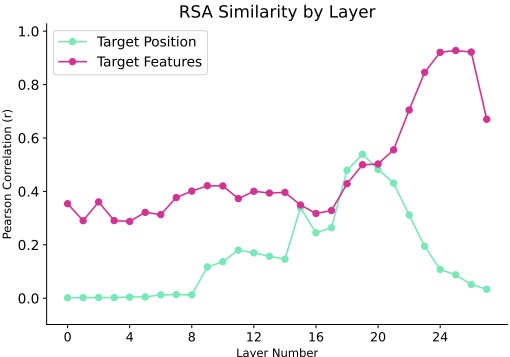

Figure 15: Representational analyses for Qwen2.5-VL-7b.

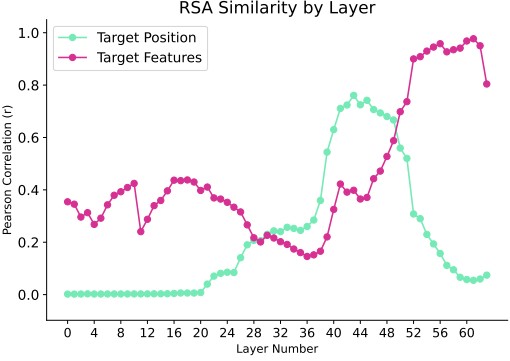

Figure 16: Representational analyses for Qwen2.5-VL-32b.

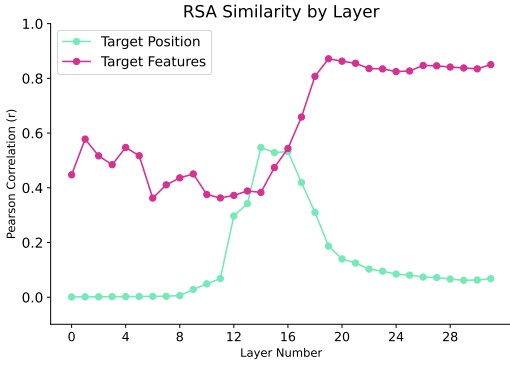

Figure 17: Representational analyses for Llava1.5-7b.

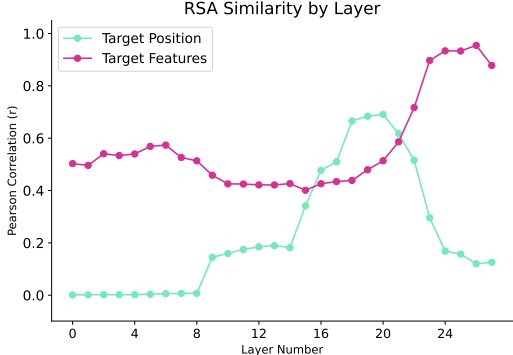

Figure 18: Representational analyses for LlavaOnevision-7b.

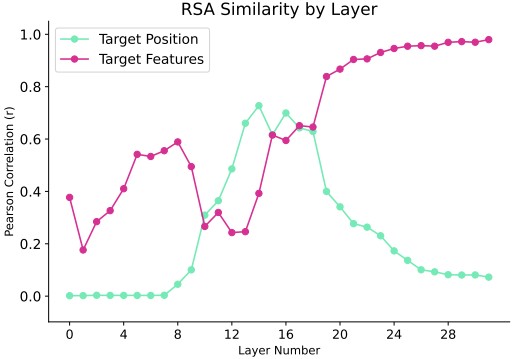

Figure 19: Representational analyses for Idefics2-8b.

## A.5.2 Causal mediation analyses

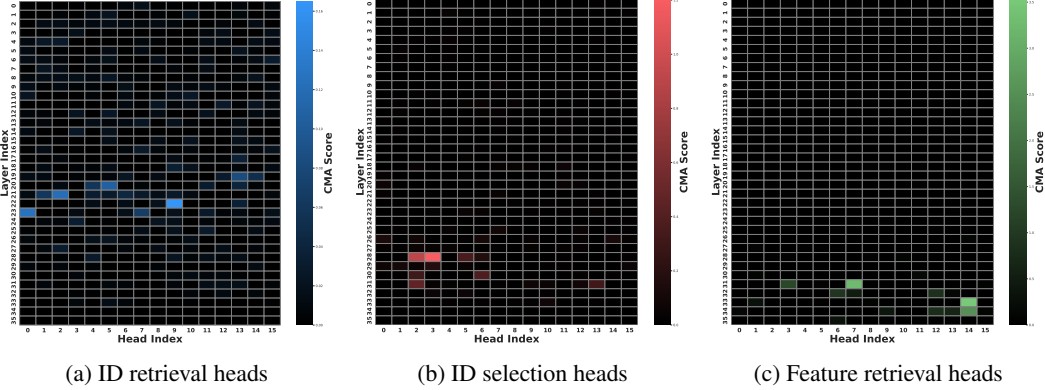

(a) ID retrieval heads      (b) ID selection heads      (c) Feature retrieval heads

Figure 20: CMA results for Qwen-2.5-VL-3b.

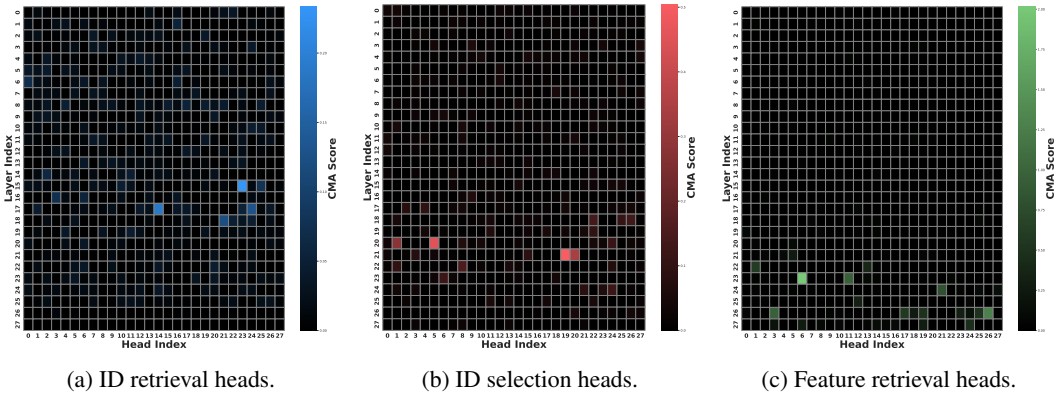

(a) ID retrieval heads.      (b) ID selection heads.      (c) Feature retrieval heads.

Figure 21: CMA results for Qwen-2.5-VL-7b.

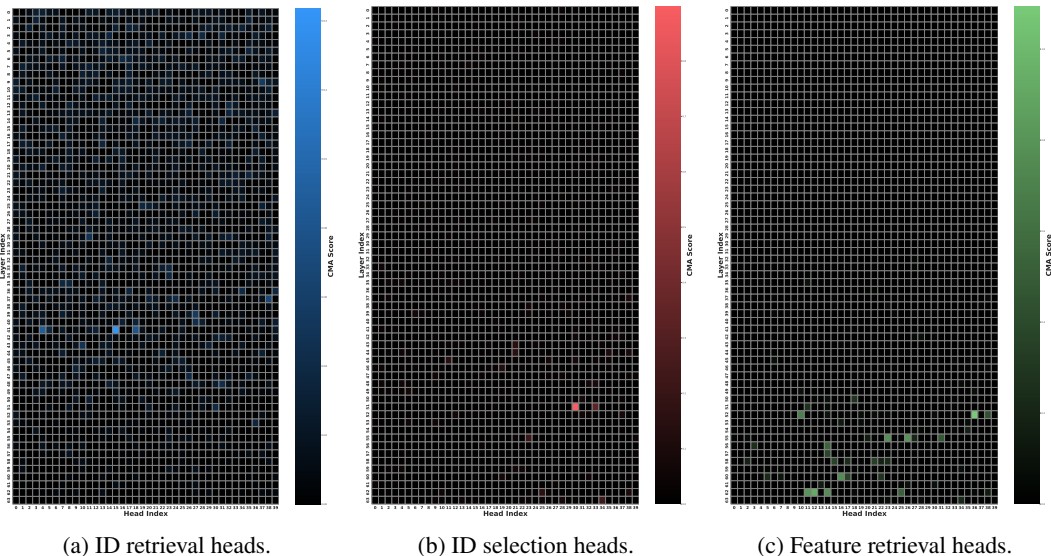

(a) ID retrieval heads.      (b) ID selection heads.      (c) Feature retrieval heads.

Figure 22: CMA results for Qwen-2.5-VL-32b.

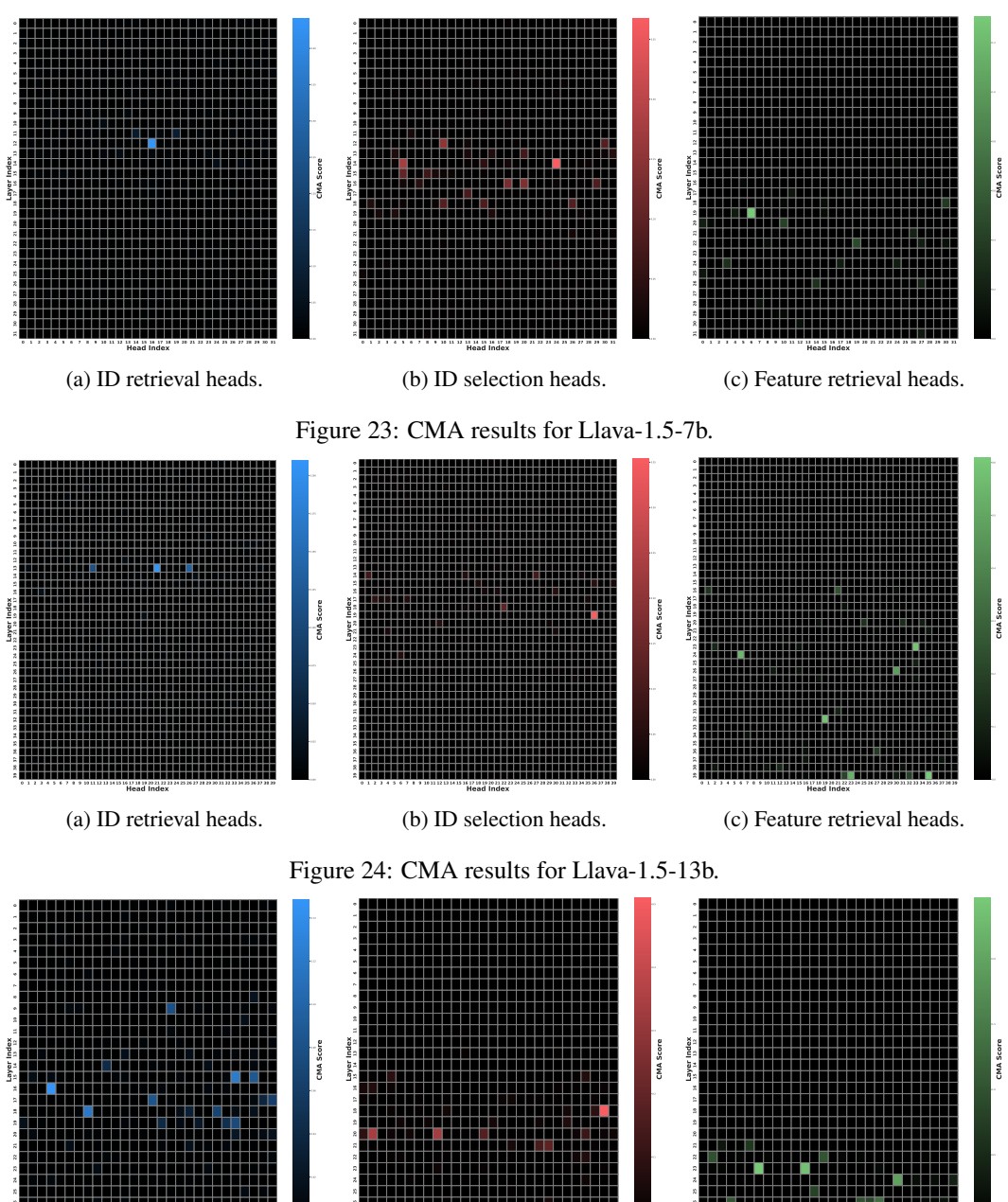

(a) ID retrieval heads.

(b) ID selection heads.

(c) Feature retrieval heads.

Figure 23: CMA results for Llava-1.5-7b.

(a) ID retrieval heads.

(b) ID selection heads.

(c) Feature retrieval heads.

Figure 24: CMA results for Llava-1.5-13b.

(a) ID retrieval heads.

(b) ID selection heads.

(c) Feature retrieval heads.

Figure 25: CMA results for Llava-OneVision.

### A.5.3 ADDITIONAL RESULTS FOR RELATIVE VS. ABSOLUTE SPATIAL CODING

**PCA Analyses**    We performed PCA analyses 26 at two levels of granularity. First, we analyzed all objects across conditions, plotting them according to either their relative grid position (Figure 26a, eg. *top left*), their absolute grid position (Figure Figure 26b) or their semantic features (Figure 26c). Second, we focused specifically on the central object (circled in dotted lines in Figure 8), coloring these representations by either relative grid condition (Figure 26d) or semantic identity (Figure 26e). If VLMs use absolute position encoding, the central object's position ID should cluster together regardless of grid arrangement, since it occupies the same absolute location across conditions. If VLMs use relative position encoding, the central object's position ID should vary systematically with the surrounding spatial context. We vizualize the PCA results for Qwen2-VL. Our PCA results showed that representations of the central object clustered primarily by grid arrangement for the Qwen2-VL 26d rather than remaining invariant across conditions. This provides evidence that some VLMs like Qwen2-VL employ a relative position encoding scheme, where spatial indices are computed based on local spatial relationships between objects rather than absolute grid coordinates.

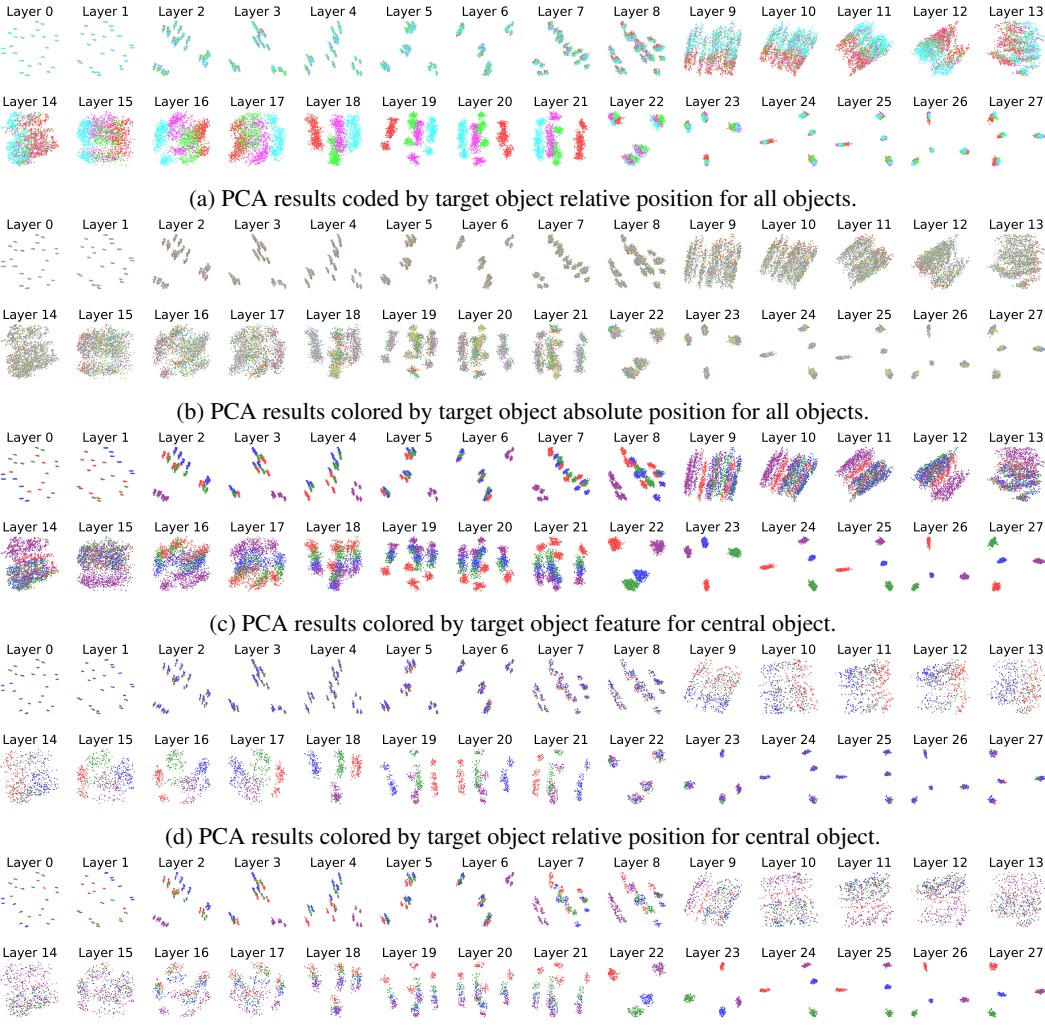

(a) PCA results coded by target object relative position for all objects.

(b) PCA results colored by target object absolute position for all objects.

(c) PCA results colored by target object feature for central object.

(d) PCA results colored by target object relative position for central object.

(e) PCA results colored by target object features for central object.

Figure 26: **Principal Component Analyses.** Stimulus condition shown in Figure 8. PCA results for all objects. (e-g) PCA results for central object only. Hidden state embeddings at last token position projected onto the top 2 principal components, coded by relative position, absolute position, or object features.

**RSA Analyses.** We performed RSA on the last token in a scene description task. The task involved completing a caption that described all but one of the objects present in the image. Our results (Figures 27 and 28) confirmed that the Qwen family of models employ a relative position coding scheme, while the difference was less pronounced for the Llava family. Interestingly, those results are also reflected in the effectiveness of the interventions reported in Figure 30. We also tested a model from the Idefics family, finding similar results as those seen for the Qwen models (Figure 29).

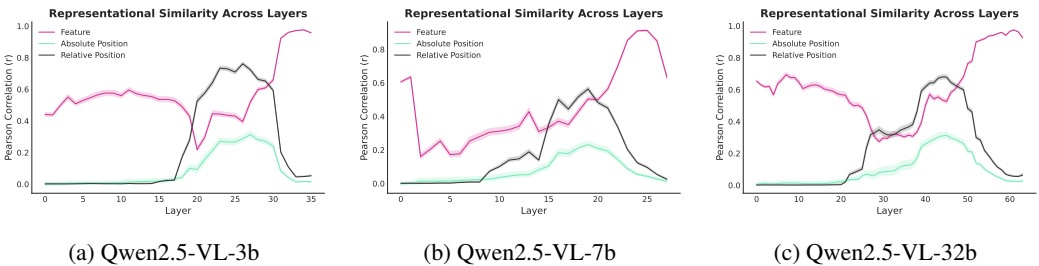

(a) Qwen2.5-VL-3b     (b) Qwen2.5-VL-7b     (c) Qwen2.5-VL-32b

Figure 27: RSA results comparing relative vs. absolute spatial coding schemes for Qwen2.5 models.

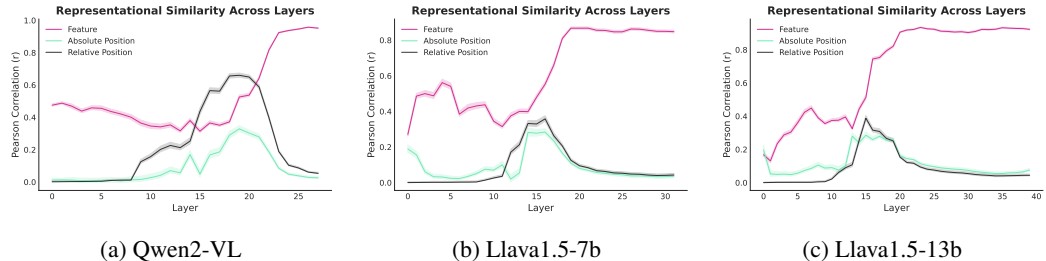

(a) Qwen2-VL     (b) Llava1.5-7b     (c) Llava1.5-13b

Figure 28: RSA results comparing relative vs. absolute spatial coding schemes for Qwen2 and Llava1.5 models.

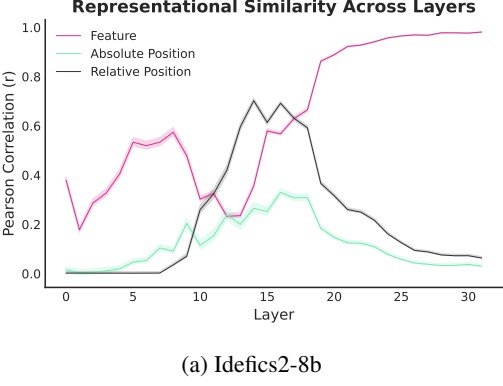

(a) Idefics2-8b

Figure 29: RSA results comparing relative vs. absolute spatial coding schemes for Idefics2-8b.

**Intervention Setup.** Figure 30 shows the results of the intervention testing for relative vs. absolute spatial coding, as described in Section A.4.

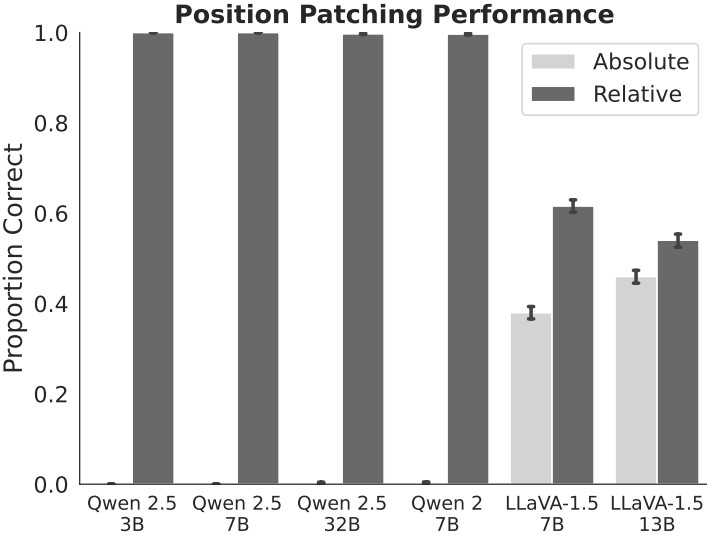

Figure 30: Intervention results testing for relative vs. absolute spatial coding schemes.

### A.5.4 BINDING ERRORS

Here we report RSA results in the low vs. high entropy conditions, illustrating the relationship between position IDs and binding errors (similar to Figure 6), for all models.

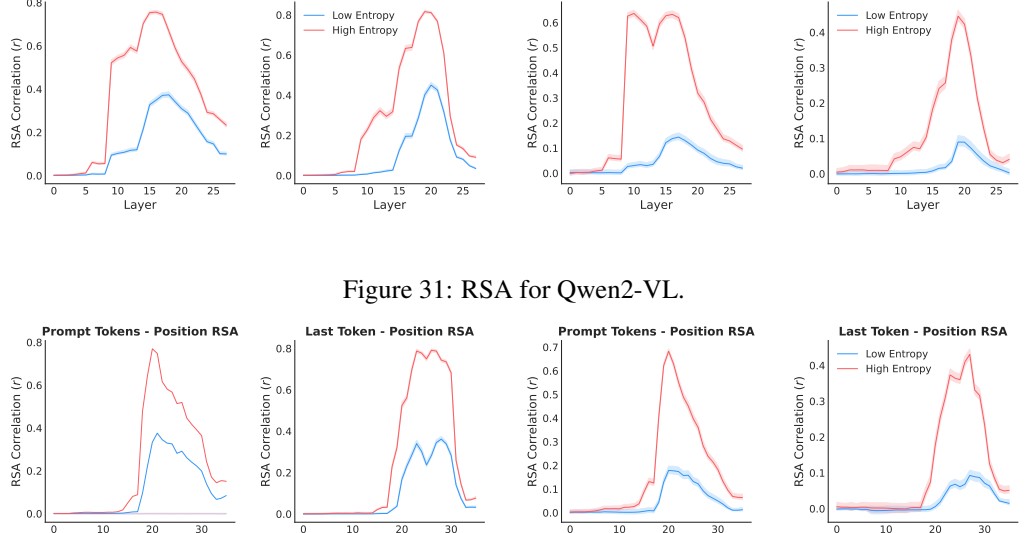

Figure 31: RSA for Qwen2-VL.

Figure 32: RSA for Qwen2.5-VL-3b.

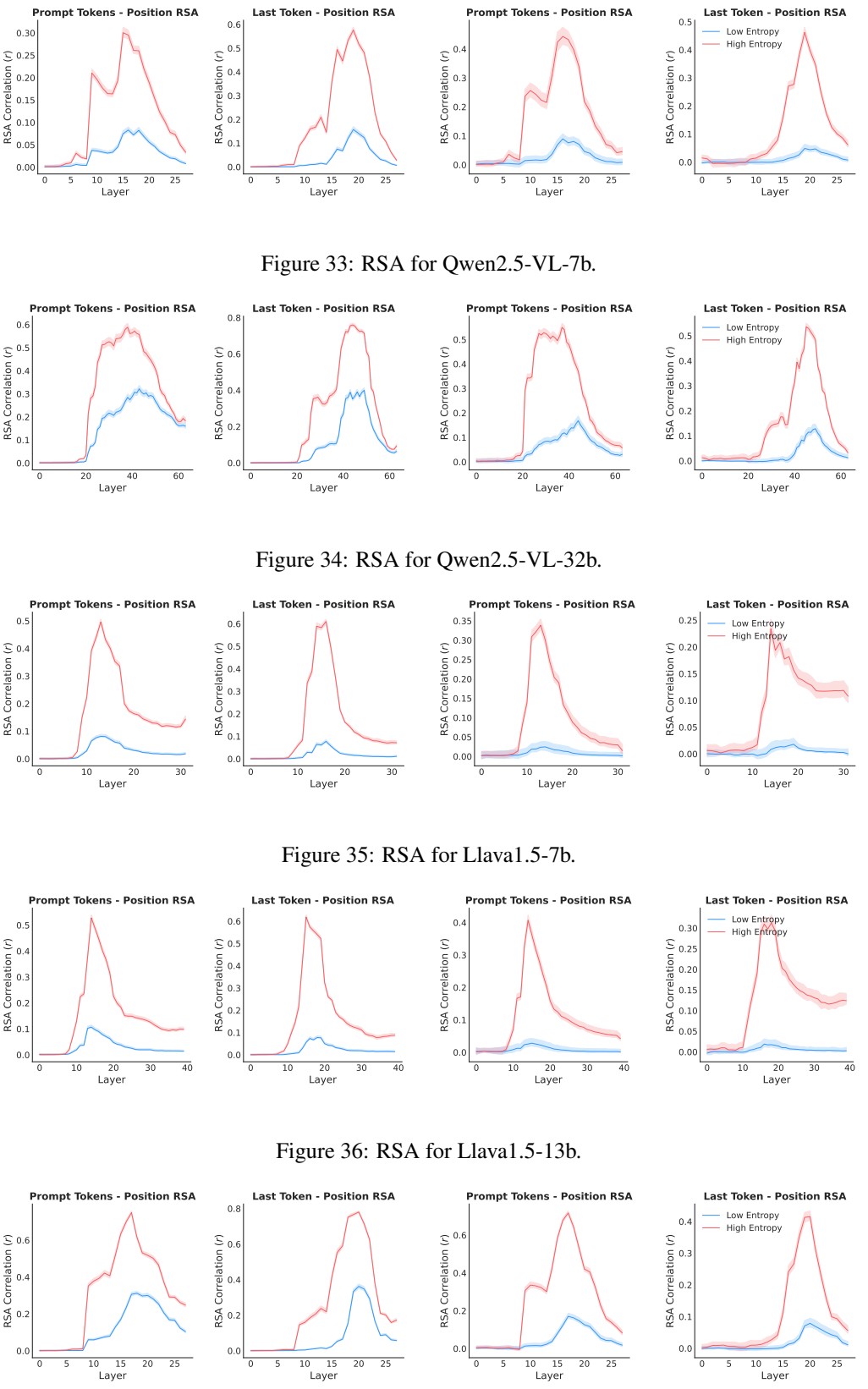

Figure 33: RSA for Qwen2.5-VL-7b.

Figure 34: RSA for Qwen2.5-VL-32b.

Figure 35: RSA for Llava1.5-7b.

Figure 36: RSA for Llava1.5-13b.

Figure 37: RSA for LlavaOne-7b.

### A.5.5 OPTIMIZING HYPERPARAMETERS FOR POSITION ID INTERVENTION

In this section, we report the hyperparameter sweep results for the position ID intervention using the PUG environment. We report intervention performance as a function of the number of top-K heads that are intervened on and the magnitude of the intervention coefficient $\alpha$, as described in Section 4.2.

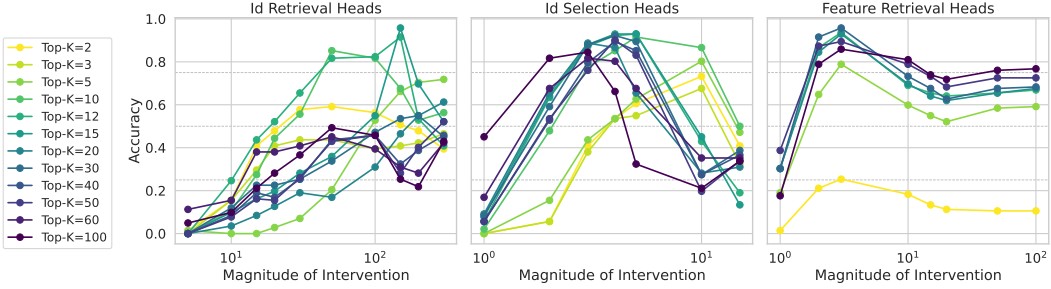

Figure 38: Qwen-2.5-VL-3B

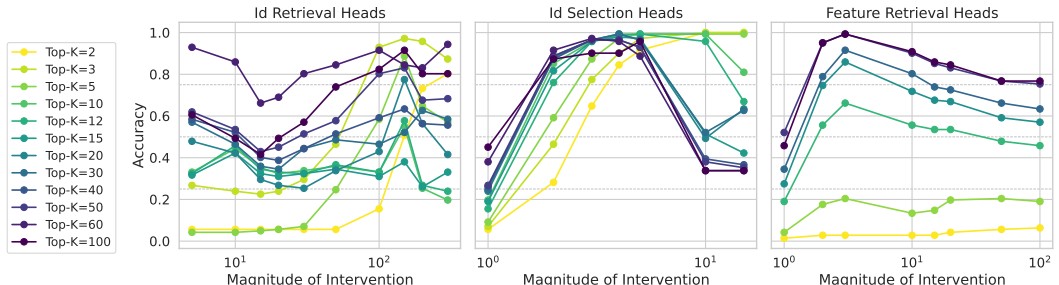

Figure 39: Qwen-2.5-VL-7B

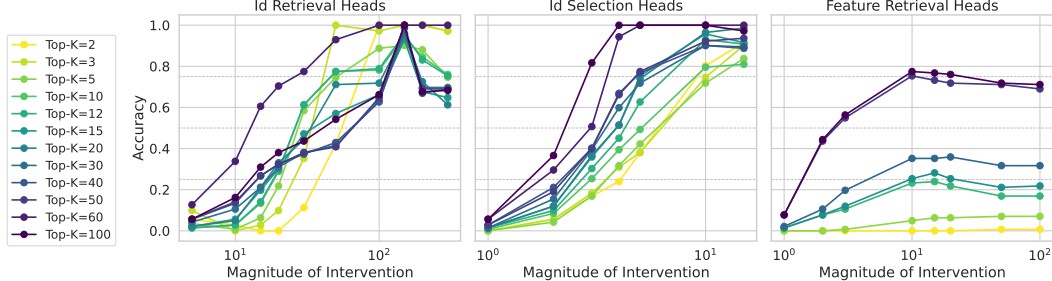

Figure 40: Qwen-2.5-VL-32B

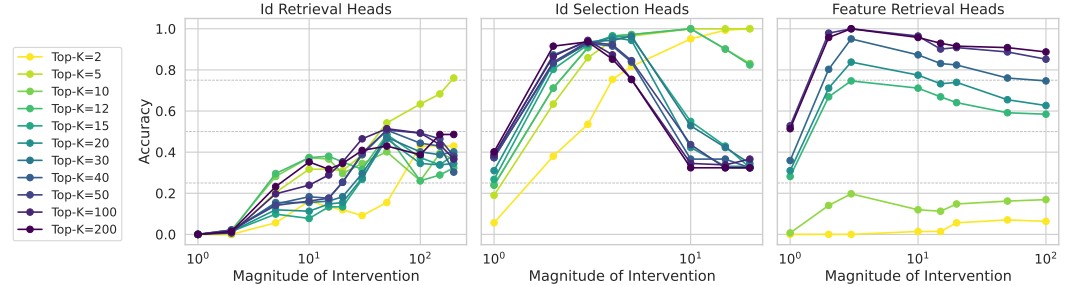

Figure 41: Qwen-2-VL

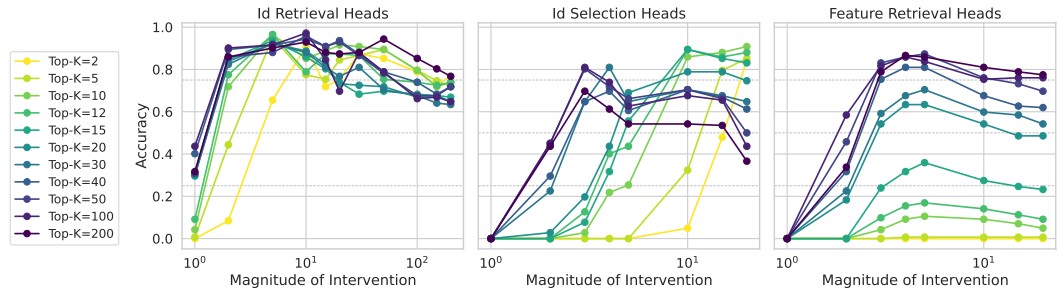

Figure 42: Llava-1.5-7B

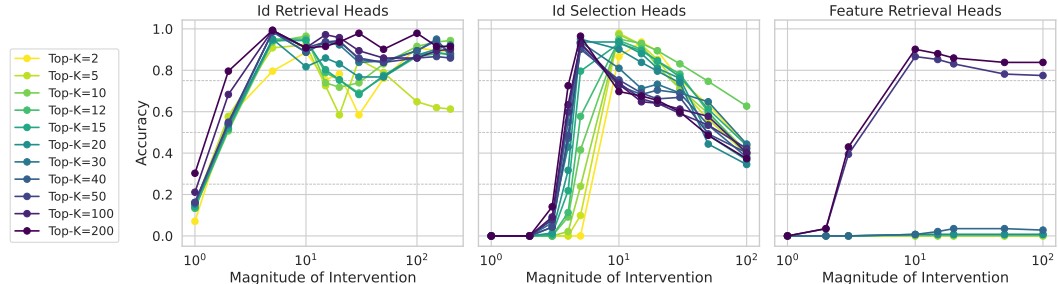

Figure 43: Llava-1.5-13B

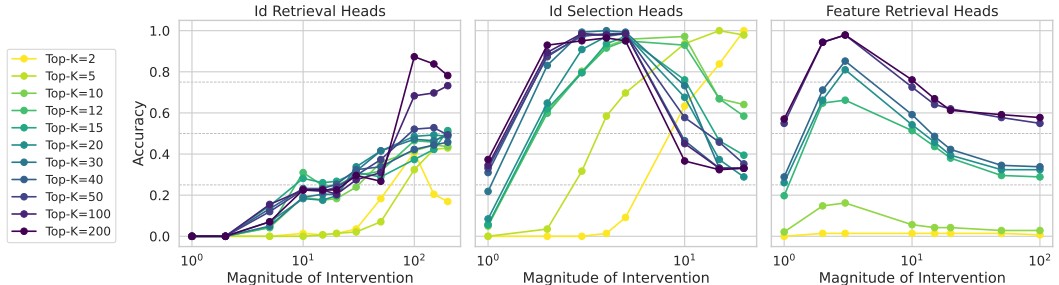

Figure 44: Llava-OneVision

# B  GENERALIZATION OF POSITION IDS TO REAL-WORLD SETTINGS

Figure 45: Illustration of the intervention performed with the COCO dataset.

In this section, we verify that the use of position IDs generalizes to real-world images. Importantly, in contrast to the experiments conducted in the main sections, for real-world spatial arrangements we do not know the assignment of IDs to objects in the image. We hypothesized that the ID assignment order would be reflected by the order in which objects were enumerated in an open-ended scene description task. Accordingly, we obtained an ordering for the objects in an image based on the order in which the model described these objects, and then used this ordering to perform an intervention (Figure 45).

Specifically, we first created separate source and target sets using images from the COCO dataset (Lin et al., 2014). The source set was used to estimate position IDs that were then used to intervene on scene description for the target set. For each image in the source set, we first obtained the model's assigned order by presenting the prompt 'In this image there is 1. a'. We then parsed the response and extracted the first two objects $(O_0, O_1)$ listed by the model (filtering out cases in which the model described the same object twice). Then, using the same image and prompt, we extracted the output of the ID selection heads from the final position (at which the model generated $O_0$). We performed this procedure for all images in the source set and averaged the resulting embeddings, yielding $\sim ID_{O_0}$, an estimate of the position ID for object $O_0$.

We then used this estimated position ID embedding to intervene on scene description with images from the target set. For each image in the target set, we again obtained the model's assigned order for the first two objects $(O_0, O_1)$. We then presented the model with the same image and the prompt 'In this image there is 1. a $O_0$ 2. a' (where $O_0$ was filled in based on the model's assigned order), and replaced the output of the ID selection heads at the last token position with the estimated position ID embedding $\sim ID_{O_0}$. Our prediction was that this intervention should cause the model to repeat $O_0$.

For this experiment, we used the COCO 2017 validation set, with 3 random splits for the source and target sets. We performed the intervention on the top 50, 100, and 200 ID selection heads (as identified by the CMA scores in A.5.2). We report the results for models from the Llava1.5 and Qwen2.5 families in Figure 46.

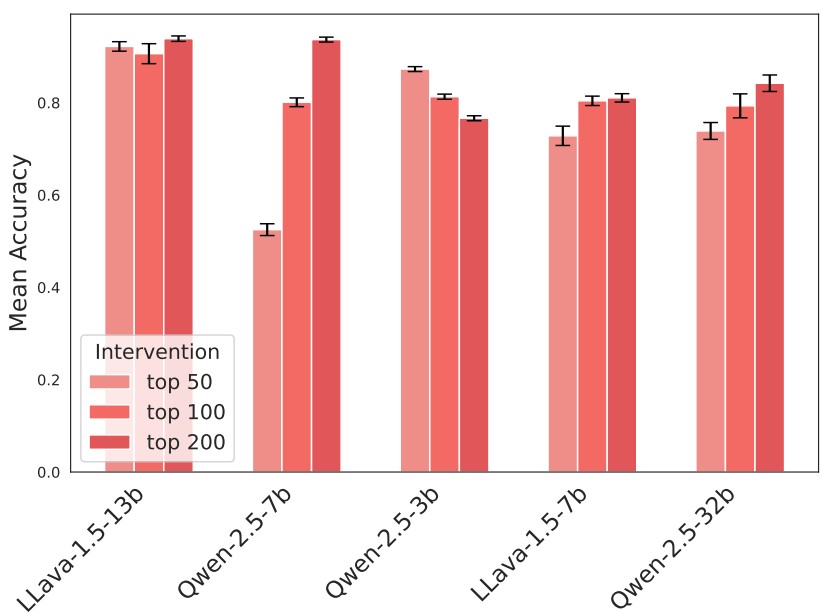

Figure 46: COCO intervention results. Intervention performed on ID selection heads.

## C INVOLVEMENT OF VISUAL SYMBOLIC MECHANISMS IN COUNTING

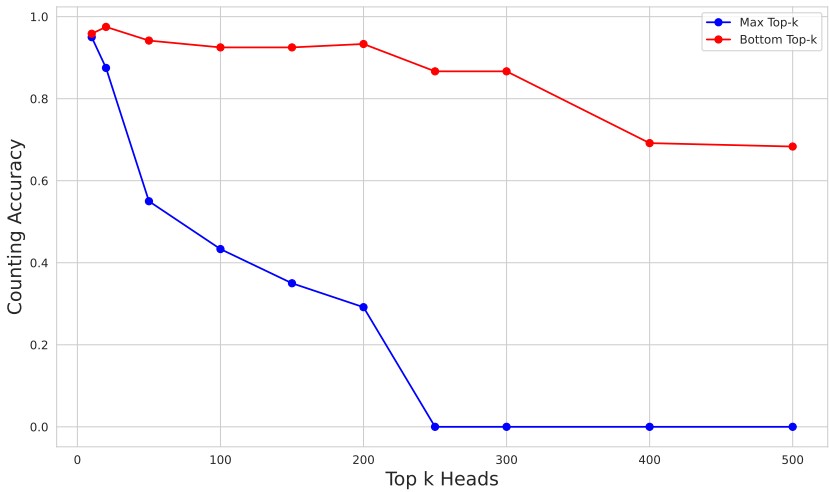

Figure 47: Ablation of top- vs. bottom-K heads (as identified by CMA) on a counting task. Note that the model contains a total of $2,560$ attention heads.

To test whether the visual symbolic mechanisms identified in the main text are involved in more complex tasks that require binding, we performed an experiment with a counting task. We hypothesized that these mechanisms are involved in enumerating the objects present in an image, and that the ablation of these mechanisms would interfere with this enumeration process, and therefore interfere with the final count provided by the model. To test this, we sorted the attention heads in the model according to their scores (as given by the maximum CMA score for each head from the 3 identified stages) and ablated (by setting the output to 0) either the top-K or bottom-K heads according to

those scores. We tested the impact of this ablation in a counting task. The task involved images with between 3 and 8 unique objects with the following characteristics:

- shapes sampled from: triangle, square, circle, cross
- colors sampled from: red, blue, green, yellow
- positions sampled randomly from a 3x3 grid

We used the following prompt for this experiment:

**Counting Task Prompt**

```
You are given an image containing multiple colored objects. Your task is
    to carefully observe the image and identify all the unique colored
    objects present.
Enumerate all the unique colored objects you find in the image, providing
    a numbered list for clarity.
After listing the objects, provide the total count of these unique
    colored objects.
Format the total count by writing 'Answer:' followed by the number. It is
    crucial to adhere to this format: 'Answer: TOTAL_COUNT'.
```

Figure 47 shows the ablation results for the best performing model on this task, Qwen2.5-32b. Ablation of the top-K heads had a much stronger impact on task performance than than the bottom-K heads, with performance falling to $0\%$ after the ablation of the highest-scoring 250 heads (out of 2,560 heads in total), whereas the model still displayed performance close to $70\%$ even after ablating the lowest-scoring 500 heads. These results indicate that the identified visual symbolic mechanisms play an important role in more complex tasks that require binding, such as counting.

## D    BINDING ERROR INTERVENTIONS

| Model | High Entropy | Low Entropy | | Improvement |
|-------|--------------|-------------|------|-------------|
| | | No Intervention | With Intervention | |
| LLaVA 1.5 7B | 70.30% | 27.50% | 32.10% | 4.60% |
| LLaVA 1.5 13B | 75.00% | 41.20% | 51.60% | 10.40% |
| Qwen 2.5-VL 3B | 91.20% | 69.00% | 80.10% | 11.10% |
| Qwen 2.5-VL 7B | 89.90% | 51.20% | 62.30% | 11.10% |

Table 1: Performance (accuracy) on high vs. low entropy scene description. Intervention performed on ID selection heads in low entropy condition.

To causally test the relationship between binding errors and failures in the position ID mechanism, we conducted an activation patching intervention targeting the ID selection heads identified in Section 3.4. We leveraged the observation that models generate robust position ID representations in high-entropy settings (distinct feature conjunctions) but suffer from representational collapse in low-entropy settings (shared features). We first computed the mean activations of the top-k ID selection heads during successful trials on the high-entropy dataset to extract a robust prototype of the spatial symbol for each grid position. We then intervened during the processing of low-entropy trials by replacing the activations of the ID selection heads with those obtained from the high-entropy trials.

The results of this intervention are presented in Table 1. Injecting position IDs from high-entropy trials significantly improved performance in low-entropy trials across all tested architectures. For instance, Qwen 2.5-VL 3B improved by 11.1%, while LLaVA 1.5 13B improved by 10.4%. Notably, the effectiveness of the intervention appears contingent on the quality of the source representations; LLaVA 1.5 7B showed the smallest improvement (+4.6%), consistent with its lower baseline performance in the high-entropy setting (70.3%). This suggests that because the source position IDs were

less precise, the resulting patch was less effective in improving performance in the low-entropy condition. Overall, these findings demonstrate that patching more precise position IDs from the high-entropy condition can attenuate binding errors during the low-entropy condition, providing causal evidence that position ID failures are responsible for binding errors.

