# OpenReview forum: "Visual symbolic mechanisms: Emergent symbol processing in Vision Language Models"
_ICLR.cc/2026/Conference — ICLR 2026 Oral_

### Official Review · Reviewer_LWd9 · 2025-10-18

**Soundness:** 4
**Presentation:** 4
**Contribution:** 3
**Rating:** 8
**Confidence:** 3

**Summary:**

This paper investigates emergent symbolic mechanisms for binding features and objects in vision language models. It presents evidence for a binding process where the model first computes a spatial index for the target object and then uses that index to retrieve its features. Overall, this paper looks like a good paper with solid methodology and evidential conclusions to me. The proposed binding mechanism is justified and intuitive, and more importantly generalisable across different models, which makes the paper relevant and useful to community.

**Strengths:**

1. The paper is well-written and easy to follow.
2. The analysis methods used including PCA and RSA to localise when position and features dominate, CMA and intervention to find contributing heads are soild methods.
3. Given that the experiments throughout 7 VLMs show a consistent pattern in the feature binding mechanism, the findings and conclusions are generalisable.

**Weaknesses:**

1. Although 7 VLMs are tested, they are from 2 model families (Qwen and Llava). The mechanism still generalises with the current setting but can be strengthened when tested on more model families.
2. The experiment is conducted under controlled synthetic settings. While this is required for a grounded analysis, explanaing how the mechanism can be exploit/identified in open-domain can be useful.
3. Some more explanation under Fig1 caption can help me to understand the paper better.

**Questions:**

The paper says improving binding performance may require either architectural innovations or training strategies. Is there a good benchmark/criterion (i.e. the one used in this paper) to show progress? How generalisable is the conclusion to more complex open-domain?

---

> ### Author Response · Authors · 2025-11-21
> **Rebuttal Answer**
>
> We would like to thank Reviewer LWd9 for their thoughtful feedback. Please find our detailed answers below.
>
> **Additional model family (Weakness 1\)**
>
> We thank the reviewer for stressing the breadth of models tested in our set of experiments. In order to cover a model that comes from a different family, we replicated the representational analysis in Idefics2-8b. We included those results in Section A.5.1, Figure 19 and Section A.5.3, Figure 28\. These results confirmed that this model showed the same general representational trend observed in other models, namely a peak in position information represented in intermediate layers, followed by a peak in feature information represented in later layers. These results also confirmed that this model showed a preference for relative spatial coding over absolute spatial coding. We believe that this additional analysis strengthens the generalizability of the identified position ID mechanisms.
>
> **Experiment with real-world images (Weakness 2, Question 2\)**
>
> We appreciate the suggestion to validate our findings on real-world data. To demonstrate that symbolic mechanisms persist amidst occlusion and clutter, we conducted an intervention analysis on the COCO dataset \[1\]. Lacking ground-truth grid references, we hypothesized that VLMs assign position IDs based on the models default enumeration order (i.e., the first generated object is assigned $ID=0$, and the second generated object is assigned $ID=1$). We tested this by extracting the activations of the ID selection heads as the model predicted the first object ($Obj\_{1}$) and patching these vectors into the model as it prepared to predict the second object ($Obj\_{2}$).This intervention successfully forced the model to revert to predicting $Obj\_{1}$ rather than $Obj\_{2}$, confirming that the position ID mechanism is a robust, general-purpose strategy deployed even in complex, naturalistic scenes. We illustrate and further detail the intervention setting in Figure 44 and Appendix section B, present results in Figure 44, and summarize the intervention effectiveness in the table below. We have also added a reference to the main text highlighting these additional results (at the end of section 4.3)
>
> |  | Qwen2.5 VL 3b | Qwen2.5 VL 7b | Qwen2.5 VL 32b | Llava1.5-7b | Llava1.5-13b |
> | :---- | :---- | :---- | :---- | :---- | :---- |
> | Best intervention accuracy | 81.0 | 93.7 | 87.3 | 81.0 | 93.8 |
>
> \[1\] Lin, T. Y., Maire, M., Belongie, S., Hays, J., Perona, P., Ramanan, D., ... & Zitnick, C. L. (2014, September). Microsoft coco: Common objects in context. In *European conference on computer vision* (pp. 740-755). Cham: Springer International Publishing.
>
> **Additional explanation for Figure 1 caption (Weakness 3\)**
>
> We have added the following additional explanation the caption for Figure 1:
>
> *Overview of position ID architecture and supporting evidence. We identify three processing stages: ID retrieval heads that retrieve the position ID of objects described in the prompt, ID selection heads that select the position ID for a target object, and feature retrieval heads that use this position ID to retrieve the features of the target object. These stages are revealed both by (b,c) representational and (d) causal mediation analyses (see text for explanation).*
>
> **Benchmarking improvements to binding mechanisms (Question 1\)**
>
> Previous work has established that VLMs suffer from deficits in tasks that require binding, such as visual search, counting, and scene description, particularly in low entropy settings (where objects share features). Performance on these tasks can therefore be considered as a useful benchmark for assessing improvements in binding mechanisms. In this work, we have shown that failures of the identified binding mechanisms are responsible for binding errors in low entropy scene description. To establish a link between these mechanisms and counting, we have also conducted a new ablation experiment (Appendix C) focusing on a counting task, where we found that ablating the identified mechanisms has a significant impact on counting performance. The identified mechanisms are therefore important for performing tasks such as low-entropy scene description and counting, and these tasks will be useful benchmarks to assess further improvements in binding mechanisms.
>
> We hope we have addressed all of the reviewer's concerns and are happy to answer any additional question the reviewer might have.

---

> > ### Comment · Reviewer_LWd9 · 2025-11-25
> >
> > Thanks for the rebuttal response. The authors have addressed most of my concerns, and I will keep my score to support the paper.

---

> > > ### Author Response · Authors · 2025-11-25
> > >
> > > We would like to thank Reviewer LWd9 for their time and for supporting our paper!

---

### Official Review · Reviewer_16UN · 2025-11-01

**Soundness:** 3
**Presentation:** 3
**Contribution:** 3
**Rating:** 6
**Confidence:** 4

**Summary:**

This paper investigates whether vision-language models employ symbolic mechanisms similar to those found in language models for solving the binding problem: the challenge of correctly associating features to form coherent object-level representations. Using a scene description task where models must describe remaining objects in multi-object images, the authors employ a thorough suite of methodologies in order to uncover the underlying mechanisms across a range of VLM architectures: PCA, representational similarity analysis (RSA), causal mediation analysis (CMA), and targeted interventions. The core finding is a three-stage binding process: 1) Position ID heads compute spatial indices for objects mentioned in the prompt; 2) ID selection heads select the position ID for the target object; and 3) Feature retrieval heads use the selected position ID to access target object features. The authors then perform a series of analyses to characterize the binding mechanisms in greater detail. They find that models tend to use relative vs. absolute position IDS; that the binding mechanisms generalize to more realistic images; that position IDs are localized to image patches containing the relevant object; and that the same position IDs are used in more complex relational reasoning tasks. They finally perform an error analysis, demonstrating that binding errors correlate with failures in position ID processing.


The authors then demonstrate that binding errors correlate with failures in position ID processing, particularly in low-entropy conditions where objects share features.

**Strengths:**

1. **Importance of the research question**. The binding problem represents a fundamental challenge in AI systems, and understanding how VLMs solve it has significant implications for improving multi-object reasoning capabilities.
2. **Novel mechanistic insight into VLMs**. The paper makes a strong conceptual and empirical contribution by identifying position IDs (content-independent, spatially grounded indices that play a symbolic role in feature binding). This is a novel mechanistic finding in multimodal interpretability.
3. **Methodological rigor**. The paper employs a comprehensive multi-method approach. The progression from RSA to CMA to direct interventions provides converging evidence for the proposed mechanisms.
4. **Cross-model validation & generalization**. The authors test their findings across multiple VLM architectures (Qwen-VL, LLaVa variants). The appendices contain detailed methodological descriptions, extensive additional analyses, and validation experiments.

**Weaknesses:**

1. **Limited exploration of real-world consequences.** While the authors link the discovered symbolic binding mechanisms to binding failures, they do not connect the mechanisms to the downstream behavioral consequences of binding failures gestured towards in the introduction (e.g. counting, visual search). Bridging this gap would strengthen the work.
2. **No investigation into learning origins.** The paper identifies what the mechanisms are but not how they arise. Do they emerge naturally from pretraining data distributions, or are they artifacts of architectural bias?
3. **Unclear how to improve binding**. Related to Weakness 2, the authors do not attempt to modify / strengthen the binding mechanisms. This would be practically useful and would also enable direct causal claims about the relationship between these mechanisms and downstream behavior.

**Questions:**

1. Do the authors have hypotheses or preliminary observations about when / how these position ID mechanisms emerge during training or what factors might influence their emergence (architecture, training data, etc.)?
2. Given the discovery of these three distinct stages (retrieval, selection, feature retrieval), could architectures be explicitly designed to separate or reinforce them, similar to slot-attention or object-centric models?
3. Is there evidence that the position IDs interact systematically with the text tokens (e.g., positional embeddings in the language stream), or are they confined to the vision encoder?
4. In the scene description task, are object attributes (e.g. shape, color) repeated in the scene? (e.g. two or more green objects, two or more squares). If a scene of three objects contains three distinct colors and shapes, then it seems that models could solve the task by just picking whatever attributes are left out of the prompt rather than binding.
5. The term "symbol" is strong and implies total content-independence. To what extent do the authors believe these mechanisms are truly symbolic, e.g. in the way cognitive scientists mean?

---

> ### Author Response · Authors · 2025-11-21
> **Rebuttal Answer Part 1**
>
> We would like to thank Reviewer 16UN for their thoughtful feedback. Please find our detailed answers below.
>
> **Connection of the identified mechanisms to downstream behavior in counting tasks (Weakness 1\)**
>
> We thank the reviewer for raising this point and agree that it is important to bridge the gap between the discovered symbolic binding mechanisms and their downstream behavioral consequences. To directly address this point, we have conducted a new ablation experiment (Appendix C) focusing on a counting task. Using the best performing model (Qwen2.5-32b), we ablated either the top-K or bottom-K heads according to their CMA scores (taking the max score for each head across the 3 identified stages), and assessed the impact on a counting task. Ablation of the top-K heads had a dramatic impact on counting accuracy, falling to 0% with the ablation of the highest-scoring 250 (out 2,560) heads, whereas the ablation of the bottom-k heads had a much weaker effect on performance. These new observations provide empirical and causal validation that the visual symbolic mechanisms are necessary for performing more complex tasks such as counting. Please see Appendix C of the updated manuscript for more details on this analysis.
>
> **Factors driving the emergence of visual symbolic mechanisms (Weakness 2, Question 1\)**
>
> Thank you for raising this important question. To address this question, it will likely be necessary to perform experiments that involve training VLMs under different conditions (e.g., varying architectural features and aspects of the training data). Given the very significant computational resources needed to perform such experiments (essentially requiring reproduction of the training pipeline for large-scale models), we have chosen in this work to focus on identifying and characterizing the presence of these mechanisms in pretrained models. However, we agree that this is an interesting and important question. We can speculate on possible factors that may be important. First, several aspects of the transformer architecture may provide useful inductive biases for developing symbolic mechanisms. These include the presence of discrete tokens with shared parameters, and the use of separate embeddings for keys/queries vs. values, both of which combine to enable a form of indirection, or the use of ‘pointers’, which is an essential ingredient of symbol processing. Additionally, aspects of the training data may be important, particularly the presence of multi-object scenes and captions that describe multiple objects. We have included the following statement in the conclusion to briefly address these factors:
>
> *…it is an open question whether the emergence of these mechanisms may be driven by architectural inductive biases, such as the use of distinct query/key and value embeddings (enabling a form of indirection, or the use of \`pointers'), or distributional aspects of the training data. We leave the investigation of these questions to future work.*
>
> **Causal evidence for the link between position IDs and binding errors (Weakness 3\)**
>
> To demonstrate a causal link between the position IDs and downstream behavior, we performed a causal intervention experiment (added to Appendix D) to test whether failures of the position ID mechanism mediate binding errors. We causally intervened on the output of the ID selection heads (as identified by the causal mediation analysis), patching outputs from the easier *high entropy* condition into the more challenging *low entropy* condition, to assess whether this remedied the binding errors observed in the low entropy condition. **As shown in Table 1, this intervention yielded significant performance gains across all models** (e.g., an 11.1% accuracy increase for Qwen 2.5-VL 3B). These results provide direct causal evidence that binding errors in low-entropy settings are a downstream consequence of failures in the Position ID mechanism.
>
> **Potential improvements from object-centric architectures (Question 2\)**
>
> Our results indicate that object-centric representations (in the form of position IDs) can emerge in VLMs without strong inductive biases for object-centric processing. However, it is certainly possible that the incorporation of stronger inductive biases might further enhance the robustness of these representations, or allow them to be learned more efficiently. In particular, we expect that any architectural choice that enhances the similarity-based clustering of visual tokens should aid the development of position ID mechanisms. Such modifications could include object-centric inductive biases aimed at reducing the number of redundant image tokens such as ToME \[1\] or Q-former \[2\].
>
> \[1\] Token Merging: Your ViT But Faster, ICLR 2023
> \[2\] BLIP-2: Bootstrapping Language-Image Pre-training with Frozen Image Encoders and Large Language Models

---

> ### Author Response · Authors · 2025-11-21
> **Rebuttal Answer Part 2**
>
> **Interaction between position IDs and position embeddings of text tokens (Question 3\)**
>
> Our experiments were performed entirely on the embeddings within the LLM decoder, including those corresponding to the image patches, and those corresponding to the text tokens. Our results suggest that position IDs are represented by the visual tokens within the LLM decoder corresponding to the locations of the objects within an image (this is established by the interventions in section 4.3), though they may also be present earlier in the ViT encoder. Within the LLM decoder, the position IDs may be influenced by the structure of the position embeddings used for the decoder. It is challenging to address this question directly, as doing so would require retraining multiple VLMs from scratch under different conditions. This remains an interesting question for future work.
>
> **Viability of a bag-of-words solution (Question 4\)**
>
> We agree that a bag-of-words solution is one possible mechanism for solving the scene description task when all objects have mutually exclusive features. Thus, the ability to perform this task does not establish the presence of symbolic mechanisms. However, we designed our experiments specifically to rule out the possibility of such a bag-of-words solution. For example, in the causal mediation analyses used to identify the ID retrieval and selection heads, we patch activations from one context to another. In these experiments, *both contexts involve the same set of objects, and both contexts involve the same prompt, with the same object missing from the prompt*, e.g. as shown in Figure 10, both contexts involve an image with an orange triangle and purple triangle, and the object missing from the prompt in both contexts is the purple triangle. Therefore, a bag-of-words solution should identify a purple triangle in both contexts. This predicts that patching between the two contexts should have no effect. However, this is not what we see. We see that patching attention head outputs, particularly in intermediate layers, causes the model to repeat the first object (the orange triangle). This cannot be explained by the bag-of-words hypothesis, but is consistent with the hypothesized presence of position ID representations at intermediate layers (since the only difference between the two contexts is the position of the 2 objects). More generally, the other results presented throughout the paper, showing the representation of position information in intermediate layers that can be intervened upon to systematically alter behavior, cannot be explained by a bag-of-words strategy, even if such a strategy could hypothetically solve the task.
>
> **Purely invariant vs. approximate symbols (Question 5\)**
>
> It is an interesting question whether the identified mechanisms are truly symbolic in the sense that’s intended by some theories within cognitive science. One way to frame this question is to ask whether the position IDs are purely invariant to the features of the objects that they are bound to. The effectiveness of the causal interventions, in which, for instance, position IDs estimated from completely different objects can be used to intervene and alter model performance with 95-100% efficacy in some cases, suggests that they are largely invariant to object features, but our results do not establish that they are *purely* invariant – there may be some small amount of variance that is explained by object features. However, it is worth noting that results in cognitive science establish many cases in which human reasoning is not purely invariant but is also affected by object features \[3\]. Therefore, human cognition may also be based on approximate rather than purely invariant symbols. Establishing the precise extent of invariance in the symbolic representations employed both by these models and in the human brain is an intriguing direction for future research.
>
> \[3\] Wason, P. C. (1968). Reasoning about a rule. *Quarterly journal of experimental psychology*, *20*(3), 273-281.
>
> We hope we have addressed all of the reviewer's concerns and are happy to answer any additional questions the reviewer might have.

---

### Official Review · Reviewer_LPtx · 2025-11-05

**Soundness:** 3
**Presentation:** 3
**Contribution:** 2
**Rating:** 6
**Confidence:** 4

**Summary:**

The paper studies how vision-language models address the binding problem, which involves linking visual features, such as color and shape, to specific objects. The paper proposes that these models use spatial "position IDs" as symbolic indices to perform this binding. Through representational similarity analysis, causal mediation, and targeted interventions on various VLM families, the authors identify three groups of attention heads responsible for retrieving, selecting, and using these position IDs to recover object features. The authors argue that this mechanism explains how VLMs achieve visual binding and why binding errors occur.

**Strengths:**

The paper addresses an important question about how VLMs perform visual binding and compositional reasoning.

It evaluates a diverse range of model architectures and scales (Qwen2.5-VL, LLaVA-1.5, and LLaVA-OneVision), differing in backbones, design choices, and training data, which strengthens the generalizability of the findings.

It employs multiple complementary methods, representational analysis (RSA), causal mediation, and targeted interventions, to support its hypotheses from several analytical perspectives.

**Weaknesses:**

The main claim that VLMs develop symbolic binding mechanisms similar to those in LLMs seems incremental since comparable mechanisms are already well-established in language models. The results show that related processes emerge in multimodal settings, which seems expected given the shared Transformer backbone. Could the authors clarify what is genuinely new about these mechanisms in the visual domain beyond applying known LLM findings to spatial inputs?

The paper aims to demonstrate general mechanisms of visual binding, yet the datasets are highly controlled and simplified. Even the “photorealistic” PUG scenes contain only two clearly separated, uniformly sized objects, avoiding occlusion or clutter. Testing whether the proposed mechanisms persist under more realistic conditions would be essential to support their claim of generality.

The explanation of the relative versus absolute index analysis could be made clearer, as the underlying idea is difficult to follow in its current form.  In addition, could the observed differences between models simply reflect their positional encoding priors (for example, relative RoPE embeddings in Qwen versus absolute sinusoidal embeddings in LLaVA) rather than evidence for an emergent symbolic mechanism? Clarifying this distinction would strengthen the interpretation of the results.

Minor comments:
Line 306: Typo in “wihtin”
Line 347: Missing space.
Line 427: The reference to the Appendix does not lead to the generation details as stated. The appendix structure should be cleaned up.

**Questions:**

Could the authors clarify what is genuinely new about these mechanisms in the visual domain beyond applying known LLM findings to spatial inputs?

Testing whether the proposed mechanisms persist under more realistic conditions would be essential to support their claim of generality.

Could the observed differences between models simply reflect their positional encoding priors (for example, relative RoPE embeddings in Qwen versus absolute sinusoidal embeddings in LLaVA) rather than evidence for an emergent symbolic mechanism?

---

> ### Author Response · Authors · 2025-11-21
> **Rebuttal Answer Part 1**
>
> We would like to thank Reviewer LPtx for their thoughtful feedback. Please find our detailed answers below.
>
> **Novel contributions beyond previously identified symbolic mechanisms in LLMs (Weakness 1, Question 1)**
>
> Thank you for raising this point and providing the opportunity to clarify our contribution relative to previous work. We have added the following paragraph to the related work to address this question:
>
> *Beyond the difference in modalities (images vs. text), there are also several novel contributions of our work that go beyond the previously identified emergent symbolic mechanisms in text-only language models (Yang et al., 2025). First, although the identified mechanisms in both cases involve an emergent form of symbol processing, the specific function that these mechanisms perform is different (parsing of multi-object scenes vs. induction of abstract relational patterns). This is not merely a translation of the same circuit from the textual domain into the visual domain, but entails a different circuit altogether. Second, unlike previous results in text-only models, the emergent visual symbolic representations are distributed across multiple tokens (each object spans multiple patches), demonstrating that emergent symbol processing extends to more naturalistic and high-dimensional domains. Third, the identified visual symbolic mechanisms are directly related to a significant limitation faced by current VLMs (the binding problem), suggesting that these mechanisms may be of practical relevance for improving these models.*
>
> **Experiment with real-world images (Weakness 2, Question 2\)**
>
> We appreciate the suggestion to validate our findings on real-world data. To demonstrate that symbolic mechanisms persist amidst occlusion and clutter, we conducted an intervention analysis on the COCO dataset \[1\]. Lacking ground-truth grid references, we hypothesized that VLMs assign position IDs based on the models default enumeration order (i.e., the first generated object is assigned $ID=0$, and the second generated object is assigned $ID=1$). We tested this by extracting the activations of the ID selection heads as the model predicted the first object ($Obj\_{1}$) and patching these vectors into the model as it prepared to predict the second object ($Obj\_{2}$).This intervention successfully forced the model to revert to predicting $Obj\_{1}$ rather than $Obj\_{2}$, confirming that the position ID mechanism is a robust, general-purpose strategy deployed even in complex, naturalistic scenes. We illustrate and further detail the intervention setting in Figure 44 and Appendix section B, present results in Figure 44, and summarize the intervention effectiveness in the table below. We have also added a reference to the main text highlighting these additional results (at the end of section 4.3)
>
> |  | Qwen2.5 VL 3b | Qwen2.5 VL 7b | Qwen2.5 VL 32b | Llava1.5-7b | Llava1.5-13b |
> | :---- | :---- | :---- | :---- | :---- | :---- |
> | Best intervention accuracy | 81.0 | 93.7 | 87.3 | 81.0 | 93.8 |
>
> \[1\] Lin, T. Y., Maire, M., Belongie, S., Hays, J., Perona, P., Ramanan, D., ... & Zitnick, C. L. (2014, September). Microsoft coco: Common objects in context. In *European conference on computer vision* (pp. 740-755). Cham: Springer International Publishing.
>
> **Clarification of tests for relative vs. absolute spatial coding (Weakness 3\)**
>
> To clarify the proposed relative vs. absolute spatial coding hypotheses, and the experiments performed to test them, we have presented additional description and an illustration in the Appendix section A.4 and Figure 13\.

---

> ### Author Response · Authors · 2025-11-21
> **Rebuttal Answer Part 2**
>
> **Are the identified mechanisms truly emergent? (Weakness 4, Question 3\)**
>
> Thank you for raising this question. First, we should clarify that both Llava and Qwen use RoPE embeddings in the LLM decoder (1D RoPE embeddings for Llava and 2D RoPE embeddings for Qwen), while the ViT encoder uses absolute sinusoidal embeddings for Llava (inherited from CLIP) and RoPE embeddings for Qwen. Nevertheless, we agree that differences in the structure of the position IDs between models may be influenced by the structure of these model’s position embeddings, though it may also be driven by the disparity in training data (Qwen models are trained on significantly more multimodal data than Llava models). We have added a note addressing this possibility to the end of section 4.1:
>
> *This may be a result of differences in position embeddings for the Llava models, or may be due to Llava's significantly smaller training set.*
>
> However, it is important to emphasize that these embeddings **cannot explain the presence or structure of the identified mechanisms.** First, as shown by our intervention experiments (Figure 28), even for Llava models the relative position intervention is more effective. Second, the identified mechanisms require the model to implement several operations that are not entailed by position embeddings alone, none of which are built into the model directly, including:
>
> 1. We use images with objects that span multiple patches. Thus, the model must group position embeddings for the patches corresponding to the same object.
> 2. Each of the identified attention heads performs a distinct function, mediated by the content of their keys, queries and values. For instance, the ID retrieval heads have queries/keys corresponding to the feature subspace and values corresponding to the position ID subspace, whereas the feature retrieval heads have queries corresponding to the ID subspace and values corresponding to the feature subspace. These specific aspects of the functions performed by these attention heads are in no way built into the model, and therefore must be an emergent consequence of learning.
>
> We have added the following passage to the conclusion discussing these issues:
>
> *It is worth considering the extent to which the identified mechanisms are truly emergent, particularly given the potential relationship between a model's innate position embeddings and the structure of the position IDs. It is important to emphasize that these mechanisms involve several operations that are not entailed by position embeddings alone, and which are not built into the architecture, including the clustering of position embeddings for patches belonging to the same object, and the specific function of the three identified attention heads (mediated by the content of their queries, keys, and values).*
>
> A related, but distinct, issue is the role that *inductive biases* play in the emergence of these mechanisms. We have also added some discussion of the possibility that inductive biases may drive the emergence of these mechanisms (which is distinct from the question of whether they are literally built into the model):
>
> *…it is an open question whether the emergence of these mechanisms may be driven by architectural inductive biases, such as the use of distinct query/key and value embeddings (enabling a form of indirection, or the use of \`pointers'), or distributional aspects of the training data. We leave the investigation of these questions to future work.*
>
> **Typos and organization**
>
> We have corrected the identified typos and references, and have reformatted the appendix.
>
> We hope we have addressed all of the reviewer's concerns and are happy to answer any additional questions the reviewer might have.

---

> > ### Comment · Reviewer_LPtx · 2025-11-26
> > **thanks for clarifications**
> >
> > Thank you for the clarifications. These additions have substantially strengthened the paper, and I have no remaining concerns. I will update my score accordingly.

---

> > > ### Author Response · Authors · 2025-11-27
> > >
> > > We would like to thank again Reviewer LPtx for their time and for supporting our paper!

---

### Official Review · Reviewer_7kgv · 2025-11-06

**Soundness:** 3
**Presentation:** 2
**Contribution:** 3
**Rating:** 6
**Confidence:** 4

**Summary:**

In this paper the authors shed light on how VLMs work, specifically how they solve the binding problem (associating features together to represent distinct objects). The authors identify the “position IDs” mechanism composed of 3 steps:  first retrieval of the position ID of the image objects corresponding to the input prompt. Then, selecting the position ID for the target object, and finally retrieving the semantic features of the target object. The authors provide evidence for this scheme across 7 models using synthetic and photorealistic datasets.

**Strengths:**

- The authors identified an interesting internal working mechanism of VLMs, explaining also why VLMs sometimes fail in spatial reasoning. I think understanding how VLMs work is a high-impact problem given recent benchmarks showing that VLMs fail in spatial reasoning tasks.
- I think the authors provide enough evidence to support the “position IDs” mechanism. They identify what layers correlate with each step and perform causal mediation analysis to identify the specific attention heads that are causally involved in the 3 steps.
- The authors show that binding errors are correlated with failures in the position ID mechanism, potentially informing future improvements to VLMs architectures.

**Weaknesses:**

- It is not clear what the novelty is in the paper in terms of methodology and techniques compared to the paper of Yang et. al (2025) that identifies similar mechanisms yet for LLMs. I think the authors should discuss it in the paper.
- The correlation between VLMs failures and position ID mechanism failures is just correlation, not causation. It is not clear if mechanism failures really cause binding errors.
- I think the writing can be improved, e.g. provide the specific prompt that is used in each case, specifically in section 3.4 and in 4.1.
- Minor: there are 2 “Figure ??” in the paper.

**Questions:**

How does the size of the model affect the results ?

---

> ### Author Response · Authors · 2025-11-21
> **Rebuttal Answer**
>
> We would like to thank Reviewer 7kgv for their thoughtful feedback. Please find our detailed answers below.
>
> **Novel contributions beyond previously identified symbolic mechanisms in LLMs (Weakness 1)**
>
> Thank you for raising this point and providing the opportunity to clarify our contribution relative to previous work. We have added the following paragraph to the related work to address this question:
>
> *Beyond the difference in modalities (images vs. text), there are also several novel contributions of our work that go beyond the previously identified emergent symbolic mechanisms in text-only language models (Yang et al., 2025). First, although the identified mechanisms in both cases involve an emergent form of symbol processing, the specific function that these mechanisms perform is different (parsing of multi-object scenes vs. induction of abstract relational patterns). This is not merely a translation of the same circuit from the textual domain into the visual domain, but entails a different circuit altogether. Second, unlike previous results in text-only models, the emergent visual symbolic representations are distributed across multiple tokens (each object spans multiple patches), demonstrating that emergent symbol processing extends to more naturalistic and high-dimensional domains. Third, the identified visual symbolic mechanisms are directly related to a significant limitation faced by current VLMs (the binding problem), suggesting that these mechanisms may be of practical relevance for improving these models.*
>
> **Causal evidence for the link between position IDs and binding errors (Weakness 2)**
>
> Thank you for noting the lack of causal evidence regarding the link between position IDs and binding errors. To address this, we performed a causal intervention experiment (added to Appendix D) to test whether failures of the position ID mechanism mediate binding errors. We causally intervened on the output of the ID selection heads (as identified by the causal mediation analysis), patching outputs from the easier high entropy condition into the more challenging low entropy condition, to assess whether this remedied the binding errors observed in the low entropy condition. As shown in Table 1, this intervention yielded significant performance gains across all models (e.g., an 11.1% accuracy increase for Qwen 2.5-VL 3B). These results provide direct causal evidence that binding errors in low-entropy settings are a downstream consequence of failures in the Position ID mechanism.
>
> **Adding prompts to paper (Weakness 3)**
>
> We have included the specific prompts used to the paper (the prompt for the task discussed in section 3.4 is shown in the Appendix section A.3; the prompts for the task discussed in section 4.1 are shown in the Appendix section A.4)
>
> **Corrected figure references (Weakness 4)**
>
> We have corrected the figure references in the revised manuscript.
>
> **Effect of model size (Question 1)**
>
> Although we anticipated that the visual symbolic mechanisms might vary with model scale, we observed that the mechanisms are present to a roughly comparable extent at least across the model scales that we tested (3B, 7B, 13B, and 32B). Future work might explore the presence of these mechanisms over a broader range of model scales, as well as the influence of other factors such as the amount of training data.
>
> We hope we have addressed all of the reviewer’s concerns and are happy to answer any additional question the reviewer might have.

---

> > ### Comment · Reviewer_7kgv · 2025-11-26
> > **Thank you**
> >
> > Thank you for your detailed answer.
> > I would like to increase my score.

---

> > > ### Author Response · Authors · 2025-11-27
> > >
> > > We would like to thank again Reviewer 7kgv for their time and for supporting our paper!

---

### Author Response · Authors · 2025-12-02
**Rebuttal summary for the AC**

We would like to thank the reviewers for their time spent reviewing our paper. We would like to provide a summary of the concerns we have addressed during the rebuttal period. We believe those additions further strengthened our contribution.

### **Rebuttal Additions**

**Causal evidence linking position ID failures to binding errors**
*Raised by reviewers: 7kgv, 16UN*
We performed a new causal intervention experiment (Appendix D)**:** ID selection head outputs were patched from an easier high entropy condition into a more challenging low entropy condition. The intervention yielded **significant performance gains** (e.g., \+11.1% accuracy), providing **direct causal evidence** that position ID mechanism failures cause binding errors.

***Generalizability to real-world images***
*Raised by reviewers: LPTx, LWd9*
We **conducted a new intervention experiment using real-world images from the COCO dataset (Appendix B, Figure 44):** Intervening on ID selection heads while the model completed a list of objects present in the scene caused the model to systematically repeat objects consistent with the position ID intervention, confirming the **Position ID mechanism is robust** even amidst occlusion and clutter.

***Exploration of downstream behavioural consequences***
*Raised by reviewers: 16UN*
**We conducted a new ablation experiment on a counting task (Appendix C):** Ablating the top-K performing heads (based on their CMA scores) had a **dramatic impact on counting accuracy** (dropping to 0%), providing causal validation that our identified ID mechanisms are **necessary for complex tasks** like counting.

***Model family coverage (Qwen and Llava family only)***
*Raised by reviewers: LWd9*
**We replicated representational analysis in Idefics3-8b:** Results confirmed the same representational trend (peak in position $\\rightarrow$ peak in feature info) and the preference for **relative spatial coding**, strengthening generalizability.

***Novelty w.r.t. to prior work on emergent symbol processing***
*Raised by reviewers: 7kgv and LPtx*
We added an additional discussion in Related Work explicitly addressing the novelty and contribution of our work relative to previous work on emergent symbol processing in LLMs \[1\]. Briefly, although the mechanisms identified in both studies implement a form of symbol processing, the specific function that these mechanisms carry out is completely different (multi-object scene parsing vs. relational binding in text). Importantly, our work also demonstrates that symbolic representations can emerge for higher-dimensional, naturalistic inputs (e.g. when objects are distributed across several image patches), and also directly relates the identified mechanisms to the binding problem in VLMs, suggesting practical relevance.

\[1\] Yang, Y., Campbell, D., Huang, K., Wang, M., Cohen, J., & Webb, T. (2025). Emergent symbolic mechanisms support abstract reasoning in large language models. *arXiv preprint arXiv:2502.20332*.

***Clarification of the relative vs absolute coding scheme***
*Raised by reviewers: LPtx*
**We added more description and an illustration (Appendix A.4, Figure 13):** Clarified the proposed hypotheses and the experiments performed to test them.

### **Strengths outlined in initial reviews**
In their initial reviews, reviewers consistently outlined :
* **Importance and Impact of Research** : The paper addresses the binding problem, recognized as a fundamental challenge in AI and a high-impact problem given known VLM failures in spatial reasoning (7kgv, LPtx, 16UN).
* **Novel Mechanistic Insight** :  The identification of the three-stage Position ID mechanism (retrieval, selection, feature retrieval) is a strong conceptual and empirical contribution, offering novel mechanistic insight into VLM operation (7kgv, 16UN, LWd9).
* **Methodological Rigor** : The methodology is sound and comprehensive, employing a multi-method approach (RSA, CMA, interventions) that provides converging evidence for the hypotheses (LPtx, 16UN, LWd9, 7kgv).
* **Generalizability and Robustness** : The findings are generalisable and robust, validated across a diverse range of model architectures and scales (Qwen, LLaVA, etc.). The consistent pattern found across multiple VLMs makes the conclusions highly useful to the community (LPtx, 16UN, LWd9).

We remain available if the AC has any further clarification questions.

Sincerly,

The authors

---

### Meta-Review · Area_Chair_NxQW · 2026-01-05

**Summary:**

This paper studies how vision–language models solve the binding problem, i.e., how visual attributes like color and shape are correctly associated with individual objects. It argues that this process is due to spatial “position IDs” acting as symbolic indices to perform this binding. All reviewers find this paper is clearly motivated, addressing an important task, and supported by a solid multi-method analysis. In the initial version, its main weaknesses were a lack of direct causal evidence, limited evaluation on real-world images and downstream behaviors, and unclear technical novelty compared to the LLM version of emergent symbol processing work, etc. After the rebuttal, these weaknesses are largely addressed. Some reviewers mentioned they are going to update the scores accordingly.

**Reviewer Concerns:**

Reviewers 7kgv, LPtx, and LWd9 clearly state that the rebuttal addresses their concerns. Reviewer 16UN did not have further discussion after the rebuttal. The initial concerns and questions included limited exploration of real-world consequences, missing factors driving the emergence of visual symbolic mechanisms, unclear ways to improve binding, and a few specific questions. Some of these were also raised by other reviewers. In the rebuttal, the authors provide detailed responses, for example by adding a new causal intervention experiment that shows causal evidence for the link between position IDs and binding errors. I believe these concerns and questions are largely addressed.

**Reviewer Scores:**

Reviewers 7kgv, LPtx, and LWd9 state that they will either remain positive or improve their scores after the discussion. Considering that Reviewer 16UN’s initial assessment is 6 and that the concerns are largely addressed, I think this reviewer may also raise the score to 8, if not remain at 6.

---

### Decision · Program_Chairs · 2026-01-26

Accept (Oral)